# Clinical Significance and Systematic Expression Analysis of the Thyroid Receptor Interacting Protein 13 (TRIP13) as Human Gliomas Biomarker

**DOI:** 10.3390/cancers13102338

**Published:** 2021-05-12

**Authors:** Ssu-Han Chen, Hong-Han Lin, Yao-Feng Li, Wen-Chiuan Tsai, Dueng-Yuan Hueng

**Affiliations:** 1Graduate Institute of Medical Sciences, National Defense Medical Center, Taipei 114, Taiwan; kido9300630@yahoo.com.tw; 2Department of Internal Medicine, Tri-Service General Hospital, National Defense Medical Center, Taipei 114, Taiwan; jerry24062645@gmail.com; 3Department of Pathology, Tri-Service General Hospital, National Defense Medical Center, Taipei 114, Taiwan; liyaofeng1109@gmail.com (Y.-F.L.); ab95057@hotmail.com (W.-C.T.); 4Department of Neurological Surgery, Tri-Service General Hospital, National Defense Medical Center, Taipei 114, Taiwan; 5Department of Biochemistry, National Defense Medical Center, Taipei 114, Taiwan

**Keywords:** TRIP13, IDH, glioma, GBM

## Abstract

**Simple Summary:**

High expression of TRIP13 is associated with *IDH*-wild type gliomas. Patients with elevated TRIP13 levels are indicative of the poor survival outcome. The study aimed to provide comprehensive information about the oncogenic potential of TRIP13 in clinical significance for gliomas. We found that TRIP13 co-expressed genes implicated in tumorigenesis and therapeutic resistance may regulated by TP53 and FOXM1. The aberrant expression of TRIP13 in gliomas was uncovered to be regulated by distinct underlying mechanisms, including the DNA methylation and dysfunction of miRNA targeting (such as miR-29 family). TRIP13-expressing tumors also showed high aneuploidy levels and reduced ratio of CD8^+^/Treg, which have unfavorable effects on patient outcome. These results demonstrate that TRIP13 is crucial for tumor development and reveal the emerging therapeutic potential of gliomas.

**Abstract:**

The prognosis of malignant gliomas such as glioblastoma multiforme (GBM) has remained poor due to limited therapeutic strategies. Thus, it is pivotal to determine prognostic factors for gliomas. Thyroid Receptor Interacting Protein 13 (TRIP13) was found to be overexpressed in several solid tumors, but its role and clinical significance in gliomas is still unclear. Here, we conducted a comprehensive expression analysis of TRIP13 to determine the prognostic values. Gene expression profiles of the Cancer Genome Atlas (TCGA), Chinese Glioma Genome Atlas (CGGA) and GSE16011 dataset showed increased TRIP13 expression in advanced stage and worse prognosis in *IDH*-wild type lower-grade glioma. We performed RT-PCR and Western blot to validate TRIP13 mRNA expression and protein levels in GBM cell lines. TRIP13 co-expressed genes via database screening were regulated by essential cancer-related upstream regulators (such as TP53 and FOXM1). Then, TCGA analysis revealed that more TRIP13 promoter hypomethylation was observed in GBM than in low-grade glioma. We also inferred that the upregulated TRIP13 levels in gliomas could be regulated by dysfunction of miR-29 in gliomas patient cohorts. Moreover, TRIP13-expressing tumors not only had higher aneuploidy but also tended to reduce the ratio of CD8^+^/Treg, which led to a worse survival outcome. Overall, these findings demonstrate that TRIP13 has with multiple functions in gliomas, and they may be crucial for therapeutic potential.

## 1. Introduction

Glioblastoma multiforme (GBM) is the most common type of gliomas among adults, accounting for 56.6% of all intracranial tumors [1,2]. The overall survival time is approximately 14.6 months on average despite standard treatments, including aggressive surgical resection [3]. The poor prognosis is attributed to invasive characteristics, including increased mitotic activity and chemo- and radio-resistance [4,5]. The World Health Organization (WHO) classified the pathologic grading of human gliomas [6]. High-grade gliomas have worse prognosis and elevated mortality [7]. Therefore, it is pivotal to determine prognostic factors and molecular targets for therapy in brain gliomas.

Thyroid Receptor Interacting Protein 13 (TRIP13), the so-called pachytene checkpoint 2, is an ATPase family member which was first identified as a protein interacting with the human papillomavirus (HPV) E1 protein [8]. Previous research indicated that TRIP13 not only served as a mitotic checkpoint silencing protein but also played an important role in meiotic recombination and DNA break formation [9].

The activation of TRIP13 enhances tumor proliferation and malignancy [9]. Recently, TRIP13 has been found to be overexpressed in various types of cancers, including colorectal cancer [10], lung adenocarcinoma [11], ovarian cancer [12], prostate cancer [13] and head and neck cancer [14]. However, the biological role and clinical significance of TRIP13 in human glioma remains poorly elucidated.

In this study, we screened the Cancer Genome Atlas (TCGA), Chinese Glioma Genome Atlas (CGGA) and GSE16011 datasets and found that TRIP13 mRNA expression elevated in advanced grade of gliomas and isocitrate dehydrogenase 1 wild type (*IDH1*-WT) in lower-grade glioma. Moreover, TRIP13 was negatively correlated with overall survival and progression-free survival time in different grade of glioma patients, implying TRIP13 could be an oncogenic factor in the disease. In vitro experiments demonstrated that TRIP13 mRNA expression was notably higher in glioma cells compared to normal brain tissue, and its proteins were highly produced in human glioma cell lines. The functional enrichment and constructed network analyses were also performed to explore TRIP13. Its relevant genes may be involved in several crucial cancer pathways and potentially regulated by upstream regulators. We also presented the aberrant expression of TRIP13 was associated with epigenetic regulation (such as DNA methylation) and micro-RNAs (miRNAs) targeting. Finally, the TCGA analysis revealed that TRIP13 mRNA expression was positively correlated with glioma aneuploidy and may potentially modulate immune cell populations in the microenvironment. These data provide the first comprehensive evaluation of TRIP13 in gliomas.

## 2. Results

### 2.1. Over-Expressed TRIP13 Positively Associated with Malignant Progression of Gliomas

We initially examined the mRNA expression levels of TRIP13 in normal brain and different grades of brain glioma tumor tissues using the TCGA database. This open resource allowed the standardized mRNA expression of TRIP13 in normal brain and glioma tissues to be obtained. The analysis results showed that TRIP13 expression was significantly increased in glioblastoma (Figure 1A). Then, we evaluated the relationship between expression of TRIP13 and the WHO grading system of glioma. In the TCGA database, high-grade glioma showed higher expression of TRIP13 compared with low-grade glioma (Figure 1B). These results were also confirmed in the CGGA database and GSE16011 array cohort (Figure 1C,D). Moreover, the TCGA analysis exhibited that TRIP13 expression was higher in the isocitrate dehydrogenase 1 wild type (*IDH1*-WT) than in the *IDH1* mutation (*IDH1*-MUT) in lower-grade glioma (LGG), but not significant in primary GBM (Figure 1E). The above results were validated from the CGGA database but neither differs in the LGG and GBM in GSE16011 array cohort (Figure 1F,G). According to the Human Protein Atlas database, TRIP13 is located in the nucleoplasm or in the cytoplasm where microtubules are present in A549 and U-2 OS cells (Figure 1H). Furthermore, we determined TRIP13 mRNA and protein levels in normal brain and GBM cell lines by quantitative PCR and Western blot analysis. We confirmed that TRIP13 was highly expressed in several GBM cell lines (U87MG, LN229, GBM8401, U118MG, LNZ308; Figure 1I,J) than in the normal brain cells. These findings suggested that the elevated levels of TRIP13 predicted a high malignant glioma. In addition, we investigated TRIP13 expression (by RNA-seq data) for specific tumor anatomic structure identified by H&E staining used Ivy Glioblastoma Atlas Project. Specially, high expression of TRIP13 in GBM was enriched in the microvascular proliferation (Figure 1K), which is important for the tumor progression.

### 2.2. TRIP13 Expression Is Significantly Associated with Poor Outcome in Glioma

To investigate the potential association between TRIP13 expression and the patient prognostic outcome, we performed the survival analysis in GSE16011, CGGA, and TCGA data. The GSE16011 data indicated that increased TRIP13 levels in all grade and lower-grade glioma patients were predominantly associated with poor overall survival time (OS) (Figure 2A,B). Then, we selected CGGA and TCGA data to confirm the results (Figure 2D,E,G,H). The GBM patients with high TRIP13 expression had poor outcomes in CGGA and TCGA (Figure 2F,I) (no statistical significance in GSE16011, Figure 2C). We also investigated the progression-free survival (PFS) of these groups in TCGA, and found that all grade and lower-grade glioma patients with high TRIP13 levels showed shorter PFS (Figure 2J,K), but this was not significant in GBM patients (Figure 2L). Moreover, we used the TCGA data including all grade, lower-grade, and GBM patients to analyze the correlations with clinical recurrence. The results indicated that upregulation of TRIP13 displayed higher recurrence rates in all grade and lower-grade glioma (Figure 2M,N), but this was not observed in GBM patients (Figure 2O). In spite of this, these data still suggest that TRIP13 expression is a poor prognostic factor for lower-grade glioma and GBM patients.

### 2.3. TRIP13 Is Identified as a Radiation-Responsive Gene and Upregulated in Recurrent Glioma

Glioblastoma multiforme (GBM) are malignant primary brain tumors and later inevitably develop recurrent tumors due to high rates of resistance to therapy. Initially, we were interested to explore whether TRIP13 expression would change with or without radiation therapy for LGG and GBM patients. The TCGA data showed that the mRNA levels of TRIP13 increased in response to radiation (*n* = 417), compared to those without radiotherapy (*n* = 209) for LGG samples (Figure 3A) (no statistically significant difference in GBM), suggesting that radiosensitive role for TRIP13 may be different on grades of glioma. Clinically, recurrent GBMs are characterized by their resistance to radiation. To examine the association between TRIP13 patterns and GBM recurrence among patients, we assessed the TRIP13 expression in primary and recurrence samples, respectively. The analysis showed that the recurrent samples (*n* = 16) exposed higher TRIP13 expression status than primary samples (*n* = 497) in the TCGA glioblastoma collective (Figure 3B). Glioma stem cells (GSCs) are regarded as highly relevant to recurrence after treatment in glioblastoma patients. Therefore, we utilized the GEO profile database to analyze the TRIP13 mRNA expression of glioblastoma stem-like (GS) cell lines, corresponding glioblastoma primary tumors and GS neurospheres in the GDS3885 dataset. Both in spheres and GS cell lines, the TRIP13 expression significantly upregulated compared with the corresponding primary tumors (Figure 3C).

Past studies reported that glioma stem-like cells expressing CD133 (Prominin-1, PROM1) can survive ionizing radiation, leading to recurrence [15,16]. Recently, the proliferating cell nuclear antigen–associated factor (PAF) was identified as a novel DNA damage response (DDR) gene, which is necessary for maintaining GSC survival and promoting tumorigenicity and radioresistance [17]. MKI67, the well-characterized proliferation markers, were found heavily enriched for radioresistant and stem-like tumor cells [18]. Thus, we investigated the gene expression profiling of multiple GS cell lines and primary tumors obtained from the GDS3885 dataset to examine the connection between TRIP13 and these radioresistant and proliferative genes. The correlative analysis of GS cell lines showed that TRIP13 expression correlated with the expression of MKI67, PAF and PROM1, but not in the primary tumors (Figure 3D–F). Spheres were observed high correlated with MKI67 and PAF, but there was no statistical significance with PROM1. However, our data still suggested that TRIP13 may potentially be related to the maintenance of GSCs linked to GBM recurrence.

### 2.4. Functional Annotation and Pathway Enrichment Analysis of TRIP13 Co-Expressed Genes in Glioma

To clarify the role of TRIP13 in glioma, we firstly identified the gene strongly correlated with TRIP13 across the three independent datasets (TCGA, CGGA and GSE16011) by Pearson correlation analysis (Pearson R > 0.6). The following identified genes and clinical data derived from the TCGA dataset (*n* = 319), CGGA dataset (*n* = 232) and GSE16011 (*n* = 209) were used to draw the heat maps (Figure 4A). Moreover, 121 genes (named co-expressed genes, Appendix A) were overlapped in three mentioned datasets (Figure 4B). To further investigate the essential relations among the genes, Gene Ontology (GO) and Kyoto Encyclopedia of Genes and Genomes (KEGG) pathway enrichment analyses were generated through the DAVID and KOBAS websites. Significant GO term analysis showed that the TRIP13 co-expressed genes were mainly located in the condensed chromosome, participating largely in mitotic cell cycle process, kinase activity, enzyme binding, G1/S and G2/M phase transition and regulation of double-strand break repair (Figure 4C, Appendix A). As for KEGG analysis, the cell cycle pathway, DNA replication, p53 signaling pathway, FoxO signaling pathway and DNA repair pathway (such as homologous recombination, mismatch repair, nucleotide excision repair and base excision repair) were included (Figure 4D, Appendix A). All the above results suggested transcriptome-wide effects of TRIP13 on glioma.

### 2.5. Identification of TRIP13 and Its Co-Expressed Genes Interacting Networks, Pathways and Upstream Regulators in Gliomas

To gain insight into the detailed mechanism by which TRIP13 and its co-expressed genes are involved in glioma, we selected the TCGA RNA-seq data (log_2_ fold change > 1.5, Appendix A) as a model to analyze the following properties, including the diseases and functions, canonical pathway, upstream regulators and gene networks, by the ingenuity pathway analysis (IPA). We observed that increased expression of TRIP13 co-expressed genes predominantly reflects an activation state of the biological functions associated with “Cell viability of tumor cell lines” (z-score = 4.71, *p* = 4.02 × 10^−15^), “Cell transformation” (z-score = 2.95, *p* = 3.92 × 10^−4^), “Carcinoma” (z-score = 2.23, *p* = 8.30 × 10^−11^), “Repair of tumor cell lines” (z-score = 2.19, *p* = 1.59 × 10^−6^), “DNA replication” (z-score = 2.07, *p* = 6.80 × 10^−16^), “Cytokinesis of tumor cell lines” (z-score = 1.84, *p* = 9.42 × 10^−16^) and “Mitosis” (z-score = 1.79, *p* = 4.57 × 10^−49^). In contrast, functions show a negative impact on “Cell death of tumor cell lines” (z-score = −3.96, *p* = 1.22 × 10^−18^), “Necrosis” (z-score = −4.61, *p* = 3.78 × 10^−13^), “Formation of gamma H2AX nuclear focus” (z-score = −2.62, *p* = 5.39 × 10^−8^), “Quantity of microtubules” (z-score = −2, *p* = 7.48 × 10^−6^) and “Formation of mitotic spindle” (z-score = −1.46, *p* = 1.20 × 10^−16^) (Figure 5A). The top canonical pathway activated in TRIP13 co-expressed genes was “Mitotic roles of Polo-like kinases” (*p* = 2.51 × 10^−18^). As shown in Figure 5B, several co-expressed genes are involved in this pathway, which regulates mitotic entry and DNA damage-activated checkpoints. Additional pathways significantly associated with TRIP13 expression in glioma were “G2/M DNA damage checkpoint” (*p* = 2.34 × 10^−9^), “ATM signaling” (*p* = 2.95 × 10^−7^) and “Mismatch repair in eukaryotes” (*p* = 4.79 × 10^−5^), the top canonical pathways are shown in Figure 5C and Appendix A. Next, we conducted IPA to generate networks and determine if there were common upstream regulators for TRIP13 co-expressed genes (Figure 6A). The result showed that these genes are regulated by many potential upstream regulators (Appendix A), and further enriched in TP53 (z-score − 5.35), FOXM1 (z-score + 4.88), TCF3 (z-score − 4.00), E2F3 (z-score + 3.96) and E2F1 (z-score + 4.00) pathways, known to be important for cell proliferation, cell cycle, apoptosis, DNA damage repair and cancer progression. Figure 6B supplied the positive correlation of TRIP13 and these upstream regulators from TCGA-GBM data, especially for proliferation-associated transcription factor FOXM1 (r = 0.7683) and the cell cycle regulatory transcription factor E2F1 (r = 0.6443), where these genes are reported upregulated in various cancers and implicated in the development and progression of tumors. Furthermore, KDM1A, HGF, CSF2, VEGF, CCND1, ATP7B, KRAS and let-7 microRNA were predicted to be activated or inhibited (except for KRAS, the effect is not predicted) upstream molecules that affect the expression of TRIP13 in some direct or indirect way (Appendix A).

To describe the detailed genetic interactions of TRIP13 and its co-expressed genes, we analyzed them using the GeneMANIA plugin in the Cytoscape software (Figure 7). According to the multiple properties of relationships, the network contains *TRIP13* gene, 104 co-expressed genes with 20 related predicted interacting genes and 26,187 links, among which co-expression attributes 90.31%, co-localization 4.63%, physical interactions 2.36%, pathway 1.23%, predicted interactions 1.17% and genetic interactions 0.31%. Several important genes are reported as physically interactive with *TRIP13*, such as *BUB1B* and *MAD2L1*, which are well-characterized molecules driving chromosomal instability, DNA copy number variation as well as patterns of chromosome loss and gain in tumor progression, including breast cancer [19], colorectal cancer [20] and glioma [21]. Moreover, *FOXM1* and *HMMR* genes, which promote stemness of glioma stem cells (GSC) [22,23], resistant to apoptosis induced by oxidative stress and chemotherapy [24] and regulation of tumor metastasis [25,26], are detected co-localized with *TRIP13* gene. The research for associations and connections between diverse types of regulatory mechanisms can provide us with a deeper and broader understanding about role of TRIP13 in biological and pathological process, and it allowed more experiments to verify our concepts.

### 2.6. Epigenetic Regulation of TRIP13 Expression by DNA Methylation in Glioma

Recently, DNA modification, especially methylation, has been extensively linked to alterations in gene expression involved in epigenetic regulation of cancers. Therefore, we explored the promoter DNA methylation status across the low-grade glioma (LGG) (*n* = 532) and GBM (*n* = 172) samples in TCGA data by using the MEXPRESS database [27]. The DNA methylation profiles of the *TRIP13* gene appear to be differing between low-grade glioma and GBM according to the data visualization. It was clear that glioblastoma exhibited fewer methylation levels (the CpG sites were described by probe locations) than low-grade glioma cases (Figure 8Aa). Furthermore, the significant inverse correlation was found between TRIP13 expression and DNA methylation around the transcription start site (TSS), gene body and first exon (GBM, Pearson correlation coefficient ranges from +0.261 to −0.289, *p* < 0.05; LGG, Pearson correlation coefficient ranges from +0.096 to −0.367, *p* < 0.05 or more is significant). For GBM, we observed that DNA methylation in its promoter region near the first exon of 5′ untranslated region (5′ UTR) was positively related to TRIP13 expression (Figure 8Ab, r = 0.261, *p* < 0.05), which suggested that diverse genomic regions of DNA methylation may be differentially involved in regulating TRIP13 expression in glioma.

To validate these observations, we further examined *TRIP13* methylation levels in the TCGA LGG and GBM cohort by using the LinkedOmics tool [28] with Methylation450 assay, including 473 LGG samples and 106 GBM samples. As we expected, the *TRIP13* methylation levels in GBM samples were significantly lower compared with those in LGG class (Figure 8B). To analyze the promoter DNA methylation against mRNA expression levels in these samples, the differential expression analysis of methylation status and gene expression level was performed between GBM and LGG samples, respectively. By using the Pearson correlation analysis, we found significantly inverse correlations between promoter methylation and mRNA expression of TRIP13 in these samples (GBM: r = −0.2946, *p* = 0.0172; LGG: r = −0.3029, *p* < 0.0001; Figure 8C). These results suggest that the hypomethylation of the *TRIP13* promoter region is one of the factors involved in the upregulation of its mRNA expression in low- and high-grade glioma.

Previously, we observed a statistically significant co-expression of *TRIP13* and 121 similar genes (overlapping genes) in three independent databases (TCGA, GSE and CGGA). In order to analyze whether *TRIP13* methylation levels could also affect these gene expression patterns, we initially used the TCGA LGG and GBM methylation dataset to identify the 118 genes in LGG and 14 genes in GBM (a portion of 121 gene lists), and generated a volcano plot to visualize the distribution of differentially expressed genes (DEGs), including significantly upregulated and downregulated genes associated with *TRIP13* methylation (Figure 8D,E). Results showed these genes significantly all dysregulated, suggesting this co-expressed gene network might be affected by the DNA methylation of the *TRIP13* gene. Then, we conducted gene set enrichment analysis (GSEA) to determine the biological processes modulated by *TRIP13* methylation. As shown in Figure 8F, we found that *TRIP13* methylation had a negative correlation with related processes, such as DNA replication, mismatch repair, homologous recombination, Fanconi anemia pathway and cell cycle. We further assessed the association between the methylated level of *TRIP13* with survival in LGG and GBM patients (Figure 8G,H). The results showed that LGG patients with hypermethylation predict better outcomes, but this was not observed in GBM patients, suggesting the potential tumor heterogeneities on grading and other remaining unknown mechanisms in GBM.

### 2.7. Altered TRIP13 Expression Associated with Glioma Might Influenced by Dysfunction of miR-29 Family

MicroRNAs (miRNAs) are approximately 22 nucleotides (nt) non-coding RNAs that are recognized to play vital role in the post-transcription regulation of mRNA gene expression by binding to target sites in the 3′UTRs of genes [29]. Numerous studies revealed that miRNAs are implicated in various biological processes, such as cell proliferation, cell cycle and apoptosis, as well as cancer pathogenesis. Importantly, several altered oncogenic expression associated with cancers is virtually caused by dysfunction of miRNAs [30]. Thus, we firstly utilized the three miRNA prediction databases (miRanda, PITA and LinkedOmics-miRNASeq data) to discover three common miRNAs (miR-29a, 29b and 29c, also called miR-29 family, miR-29s), identified from all databases and downregulated in low-grade glioma (Figure 9A,B). Furthermore, we examined co-expression relationships of screened miRNAs and corresponding target mRNAs using starBase. The significantly negative correlations were shown between mir-29a (r = −0.287, *p* = 2.17 × 10^−11^), mir-29b (r = −0.273, *p* = 1.94 × 10^−10^), mir-29c (r = −0.322, *p* = 4.15 × 10^−14^) and *TRIP13* in low-grade glioma (Figure 9C). In addition, our data indicated that the expression levels of miR-29 family were all markedly downregulated in glioblastoma, compared to lower-grade glioma samples (Figure 9D), which is also consistent with the results reported from several studies [31,32]. Therefore, we speculated that the reduced expression levels of miR-29s are significantly associated with the regulation of *TRIP13* in glioma.

### 2.8. TRIP13 Regulation Is Associated with Tumor Aneuploidy Level and Reduces Immune Response

Aneuploidy, defined as somatic copy number alterations (SCNAs), is characterized by the occurrence of an abnormal number of chromosomes and chromosomal segments during cell division and generates a high level of chromosomal instability (CIN) [33]. There is increasing evidence that aneuploidy is prevalent in various cancers and it is proposed to drive tumorigenesis and generally associated with resistance to chemotherapy [34,35]. In some cancer types, the degree of aneuploidy was reported to change with tumor stages [36]. The increased levels of aneuploidy are linked to proliferation pathways, whereas adversely correlated with immune responses within individual tumor types [37]. On the other hand, positive correlations between the degree of aneuploidy and enrichment for cell cycle and proliferation transcriptional signatures have been found in several analyses of tumor samples [38,39]. In this study, we were interested to investigate the relevance between TRIP13 expression and tumor aneuploidy patterns with influence on immune signaling in glioma. We applied the clinical information obtained through tumor transcriptome from the TCGA pan-cancer study in cBioportal, including samples for GBM (*n* = 592) and low-grade glioma (LGG) (*n* = 514). Here, the visualized landscape of study origin, profiled for copy number alternations, aneuploidy score (the numbers of altered chromosome arms) and overall survival were generated for customized analysis. The high copy number alterations were detected both in GBM and LGG samples in genomic data (Figure 10A). To study clinical features associated with aneuploidy, we characterized the aneuploidy score reflecting the number of chromosome copies in tumor samples and its impact on patient survival in LGG and GBM, respectively. The aneuploidy pattern is widespread across GBM and LGG samples (at least 90% of samples has detectable aneuploidy, mean aneuploidy score of 6.13), and tumors with high aneuploidy had markedly worse survival compared to low-aneuploidy patients (LGG: median survival, 1091 days for high groups versus 3156 days for low groups, GBM: median survival, 434 days for high groups versus 587 days for low groups) (Figure 10A(a,b)). These data suggest aneuploidy is frequent in glioma and correlates with poor prognosis. We then exploited the aneuploidy score of each sample and found the significant differences between GBM and LGG. In fact, virtually all GBM (99%, mean aneuploidy score of 8.2) have at least one aneuploidy event (aneuploidy score range of 1–39), in contrast to LGG (87%, mean aneuploidy score of 3.84, score range of 1–33) (Figure 10B). In the previous study, it was shown that the spindle assembly checkpoint proteins play a vital role in cell division, and may promote CIN and aneuploidy with aberrations [40]. Thus, we examined data from LGG (*n* = 507) and GBM (*n* = 153) samples in TCGA to uncover relationships between TRIP13 expression and aneuploidy levels of tumors. We found that LGG with high levels of aneuploidy showed significant elevated expression of TRIP13 (Figure 10C), and the positive correlation (r = 0.38, *p* < 10^−^^15^) between TRIP13 expression and aneuploidy score (Figure 10E), which is consistent with previous observations [41]. However, we did not observe a significant difference with respect to the correlation between TRIP13 level and aneuploidy in GBM samples (Figure 10D,E). Moreover, current research has found that tumors with high levels of SCNAs tended to display elevated expressions of cell proliferation and cell cycle markers and reduced expressions of markers for infiltrating cytotoxic immune cells (namely immune signature) [38]. This inspired us to explore whether high expression of TRIP13 in glioma (significantly high aneuploidy populations) to be implicated in abnormalities of immune system. Here, we selected all the genes whose expression appeared to correlate positively or negatively with TRIP13 expression (these genes will be called “TRIP13-negatively correlated genes”) from TCGA RNA-seq data in LGG and GBM samples. In the case of the negatively correlated genes (Pearson correlation coefficient, r < −0.3, *n* = 948 for LGG and *n* = 402 for GBM), several immune-related genes are identified in the volcano plot result based on LGG and GBM, respectively (Figure 10F(a,b)). We then performed a Reactome pathway analysis individually, and consistently, the pathways mainly overrepresented were related to immunity, including “Immune system”, “Adaptive immune system”, “Innate immune system” and “Cytokine signaling in immune system” (Figure 10G(a,b)). To illustrate this result, we generated a heat map of the top 20 genes in the immune system category (Figure 10H(a,b)). The correlation coefficient calculated for these 20 genes collectively was −0.49 (*p* = 1.8 × 10^−^^32^) in LGG and −0.31 (*p* = 5.7 × 10^−^^5^) in GBM (Figure 10G(a,b)). To further investigate the relationship between TRIP13 negatively correlated genes and corresponding immune cell types, we conducted a Pearson analysis according to Figure 10F (r < −0.3) to screen the immune-related genes in LGG (*n* = 5) and GBM (*n* = 7), which were characterized from known immune signatures, transcriptomic data in immune cell populations or published studies (Table 1 [42,43,44,45,46,47,48,49,50,51,52,53,54,55]). There were seven immune cell types (T cells, B cells, dendritic cells, macrophages, granulocyte, NK cells and monocytes) that were highly relevant with 13 identified genes. In order to assess the extent of the decrease in the expression of each immune gene for tumors with high or low TRIP13 levels, we used the RSEM value of each gene after log transformation and z-score normalization among the samples available from TCGA to determine the expression level. In Figure 10I, the percentage decreases (% dec.) in the expression of immune genes between high and low TRIP13 level LGG and GBM were calculated by the value of percentage differences in the average expression between high and low TRIP13 level tumors. We observed that the overall expression of immune genes showed marked reductions in tumors with high TRIP13 levels relative to low TRIP13 level tumors (LGG: *p* = 10^−7^ to 10^−26^, GBM: *p* = 10^−4^ to 10^−9^). Finally, to describe the relationships among *TRIP13* and mentioned various immune genes (based on Figure 10H), we constructed the gene networks by GeneMANIA (Figure 10J). The networks composed of 34 genes (*TRIP13*, 7 immune genes and 21 predicted interactors) contain 113 interactions among which co-expression attributes 53.6%, physical interactions 40.9%, co-localization 4.06%, shared protein domains 1.22% and genetic interactions 0.21%. Within the networks, notably, CD99 was reported as physically interactive with TRIP13 (edge weight 6.33). CD99 is a glycosylated transmembrane protein frequently highly expressed in malignant glioma with marked effects on the migration, invasion and metastasis of tumor cells [56]. In addition, CD99 may rearrange the actin cytoskeleton and has been reported to be involved in regulating cell differentiation, adhesion and migration of immune cells [57,58].

### 2.9. The Potential Role for TRIP13 in Regulating the Ratio of CD8^+^ to Regulatory T Cells in Lower-Grade Glioma

Recent studies have indicated that the ratio between immune cell types with a pro-tumoral (anti-inflammatory) to anti-tumoral (pro-inflammatory) effect, instead of the whole densities of immune cell types, is considered more amenable to represent either a promoted or inhibitory direction to growing tumors in the tumor microenvironment [59]. Therefore, we used the TCGA data (obtained from Figure 10A, including 160 GBM and 514 LGG samples) to calculate the ratios between gene expressions specific for different types of immune cells and immune modulators (the gene sets were derived from Tumor Immune System Interactions Database, TISDIB and numerous published literature [60,61,62,63] for comprehensive investigation), which were used to describe the distinct immune cell/modulator populations that contributed to immune activation or immune suppression. Both in LGG and GBM samples, striking differences were observed among in high TRIP13 level tumors; the ratio between mRNA levels of CD8^+^ T cell–specific genes (n = 13) to regulatory T (Treg)-specific genes (*n* = 13) was notably reduced, compared to low TRIP13 level tumors (Figure 11A,B, Appendix A). Similarly, we also found a significant difference in LGG and GBM with respect to the ratio between anti-inflammatory modulators (*n* = 13) to pro-inflammatory modulators (*n* = 11), which inhibit immune cell activity and modulate the tumor microenvironment to facilitate pro-tumor effects (Figure 11C,D, Appendix A). However, the ratio of M1 macrophage (anti-tumoral)-specific genes (*n* = 8) to M2 (pro-tumoral) macrophage-specific genes (*n* = 8) was shown to be marked lower only in LGG with high TRIP13 level, but there was no significant change in GBM (Figure 11E,F, Appendix A). To further evaluate the impact of TRIP13-regulated immunity on clinical outcome, we performed the survival analysis between the groups based on the ratio. Our data indicated that a higher ratio of CD8^+^/Treg has a beneficial effect on clinical outcome in LGG and GBM (Figure 11G,H), but we did not observe a significant difference in the ratio of anti-/pro-tumor modulators, nor the ratio of M1/M2. Based on these results, it is suggested that TRIP13 may play a potential role in the regulation of CD8^+^ and Treg cells in the tumor microenvironment between different grades of glioma.

## 3. Discussion

Glioblastoma is one of the most common and malignant glioma types in the world. It is well known that advanced glioma is difficult to treat and worse prognosis due to the high rate of recurrence and tend to induce resistance for chemoradiation therapy. Similar to other solid tumors, the development and progression of GBM are characterized by genetic abnormality and aberrant protein expression due to accumulation of chromosomal and molecular-genetic aberrations. The oncogenic properties could be classified by distinct aspects, including genomic, epigenetic, transcriptomics, proteomics and metabolomics abnormalities, which are important for cancer biomarker discovery and characteristic identification. In this study, we present the oncogene, TRIP13 to in-depth investigation of its expression and prognostic features as well as its novel correlation with aneuploidy and immune modulation in glioma microenvironment by integrative multi-omics analysis. To our knowledge, this is the first study addressing the clinical significance and systematic expression analysis of the TRIP13 as biomarker in human glioma.

We initially uncovered the prognostic features of TRIP13 by series expression analysis of glioma using numerous cohort data. *IDH* mutation in lower-grade glioma and glioblastoma predict good prognostic outcomes compared to wild type *IDH* [64]. Our study showed that expression of TRIP13 is higher in wild type *IDH* than in mutant-type *IDH* in lower-grade glioma. The statistically significant of relationships between TRIP13 and *IDH1* phenotype in primary glioblastoma was not observed in our analysis. Several studies showed that *IDH1* mutation are very frequent in secondary glioblastoma (>80%) but very rare in primary glioblastoma (<5%) [65,66]. *IDH1* mutation is a definitive diagnostic molecular marker in secondary glioblastoma compared to primary glioblastoma [67]. Hence, the study was limited for TRIP13 expression analysis in primary glioblastoma to explore the relationship with IDH1. We also found that the mRNA expression levels of TRIP13 were elevated in higher–grade glioma patients in microarray data (also validated by mRNA and protein assay) and demonstrated significantly poor prognosis in LGG and GBM patients from TCGA and CGGA data. The TCGA data also present that upregulation of TRIP13 displayed higher recurrence rates, shorter overall survival and progression-free survival. The above findings support the potential oncogenic role of TRIP13 in human gliomas, and may serve as an indicator for patient with *IDH* mutations.

Radiotherapy is the common basis of GBM standard treatment. However, the inevitable recurrence acquired during the treatment is attributed to the ineffective response to radiation of GBM stem cells (GSCs) and initiating cells (GICs) through preferential activation of the DNA damage checkpoint response and the increased DNA repair capacity [16,68]. CD133 (Prominin-1, PROM1), a well-known marker for both neural stem cells and brain cancer stem cells [69], has been found to be enriched after radiation in gliomas. The fraction of CD133-positive glioma cells confers radio-resistance by activating DNA damage checkpoint proteins (including Chk1 and Chk2) to enhance the repair capacity for radiation-induced DNA damage. Hence, the glioma cells expressing CD133 could be the source of tumor recurrence after radiation [15,16]. In this study, we demonstrated for the first time that TRIP13 may be potentially related to recurrence of glioma in patients. We observed the up-regulated TRIP13 expression after irradiation in LGG as well as the recurrence of GBM compared to primary tumor. GBM stem cell lines (GS lines) and neurospheres displayed significantly higher expression levels of TRIP13 than those corresponding primary tumors. Additionally, the correlation analysis indicated that TRIP13 showed a markedly positive association with the Chk1 and Chk2 in GS cell lines and neurospheres compared with the primary tumors (Appendix A). Previous research has shown that GSCs develop into radioresistant accompanying the overexpression of proliferating cell nuclear antigen (PCNA)-associated factor (PAF) [17]. PAF is a crucial DNA damage-regulated factor that mediates both DNA replication and accessibility of (translesion synthesis) TLS enzymes to PCNA, thus working to bypass the DNA damage after GSC with radiation treatment [70]. The process of damage bypass induces the reassociation of PAF with PCNA, resulting in the release of TLS Pol η and the restoration of error-free DNA synthesis, thus supporting GSC self-renewal and radioresistance [17]. Here, the correlative analyses were conducted and they showed that TRIP13 and the two aforementioned radioresistant molecules (PROM1 and PAF) are significantly positively related in GS lines, but not in the relevant primary tumors. Antigen Ki-67, also known as MKI67, is a cell-proliferative protein that encodes by the *MKI67* gene and is widely used as a prognostic marker for several tumors [71,72], including glioma [73]. We also found that TRIP13 was strongly positively correlated with MKI67 in GS lines, but was not significant in primary tumors. Collectively, our analysis implied that TRIP13 may play a potential role in tumor maintenance linked to GBM recurrence.

The genes co-expressed with TRIP13 identified using the three glioma related datasets were subjected to functional and pathway enrichment analyses. GO term and KEGG pathway indicated that these genes were mainly manifested in cell cycle, DNA repair regulation and cancer-related pathways (such as p53 signaling pathway). Previous reports demonstrated that TRIP13 may directly regulate cell cycle progression, tumorigenesis, invasion and chemoresistance in other cancer types. In this regard, we further used IPA analysis to find that the activation of TRIP13-associated genes is significantly involved in “cell viability (and repair) of tumor cells”, with inhibitory effects on functions related to “cell death of tumor cell”, “formation of gamma H2AX nuclear focus” and “Necrosis”. The top canonical pathway activated in these genes was “Mitotic roles of Polo-like kinases” (*p* = 2.51 × 10^−18^), which regulate the mitotic entry and spindle pole functions as well as DNA damage checkpoint responses [74]. Moreover, Polo-like kinases (Plks) mediate the self-renewal versus differentiation process of neural progenitors by regulating the orientation of mitotic spindles [75]. The best characterized member of the Plk family, Plk1, has been found to be overexpressed in glioma [76] and reported to play an antiapoptotic role through the interaction with the tumor suppressor protein p53 [77]. Furthermore, Plk1 inhibitors are being investigated as an emerging anticancer therapy in clinical studies [78]. Indeed, we found a close correlation between TRIP13 and Plk1 (r = 0.7016, *p* < 10^−15^, Appendix A) in GBM, which provides for the rational development of new targeted therapies for glioma. TRIP13 and polo-like kinase inhibitors could be designed selectively for the drug regimens. Additional pathways significantly associated with TRIP13 expression in glioma included “G2/M DNA damage checkpoint” (*p* = 2.34 × 10^−9^) and “ATM signaling” (*p* = 2.95 × 10^−7^). Also of note is whether ATM signaling or TRIP13 are both significant to DNA damage repair [79]. We also determined upstream regulators of TRIP13 as well as co-expressed genes, such as TP53 and FOXM1, which are concordant with other studies, highlighting an important role for these molecules in cancer development [25,80]. Therefore, we inferred that TRIP13 and its co-expressed genes jointly regulate glioma tumorigenesis and progression through a complicated regulatory network.

Previous research carried out the integrative analysis of the DNA methylation profiles in different tumors with *IDH* mutations, including acute myeloid leukemia (AML) and low-grade GBM [81]. The mutant IDH status is associated with DNA hypermethylation phenotype, which provides clues about the molecular mechanisms affecting DNA methylation [82]. Our findings support these published results. In this study, TRIP13 methylation patterns are frequently altered in low-grade glioma and glioblastoma. Stratification of patient tumors based on the malignancy of gliomas informed us that the higher level of methylation of TRIP13 was identified in lower-grade glioma and had better outcomes as well, possibly attributed to the downregulation of DNA repair capacity and replication of tumor cells as a result of KEGG pathway enriched analysis. In the past decades, microRNAs (miRNAs) have been demonstrated to be extensively dysregulated in cancers through the downregulated or upregulated expression, which determined their function as oncogenic miRNA or tumor-suppressive miRNA. Recently, miRNA-29 family (miR-29a/b/c) was proven as proliferation, invasion and migration suppressors for gliomas by targeting cell division cycle 42 (CDC42) [32] and Sterol regulatory element binding protein 1 (SREBP-1)/SREBP cleavage-activating protein (SCAP) in animal models [83]. However, miR-29s are shown to be down-expressed widely in cancers and closely correlating with more aggressive phenotype [84,85], which is consistent with our results regarding the significantly decreased expression in higher-grade glioma (GBM). The inverse correlation between the expressions of TRIP13 and miR-29a/b/c was observed in our study, implying that downregulation of miR-29s may lead to over-expressed TRIP13 in gliomas. On the other hand, circRNAs are known to act as miRNA sponges to regulate the function of miRNA in the development and progression of cancers [86]. The network by 10 circRNAs which were predicted to sponge miR-29s identified by glioma samples in human circRNA database, as well as miR-29s and TRIP13, were firstly built in this study (Appendix A). In the future, we will employ further experiments for validation.

Cell aneuploidy, which is caused by the aberrations of whole chromosomes, was identified as the distinct characteristic in tumor development [87] and associated with treatment resistance and altered immune recognition [38,39]. Aneuploidy is associated with defects of the spindle assembly checkpoint (SAC), which is a signaling cascade that ensures the proper segregation of chromosomes during mitosis [88]. Previous studies have reported that TRIP13 is a novel component of the SAC pathway [89] and demonstrated to be the top-ranked genes related to chromosome instability (CIN) in several cancers [90,91]. To our knowledge, this is the first study to uncover the potential association between TRIP13 and aneuploidy in glioma. The tumor transcriptome analysis of TCGA indicated that glioblastoma displays chromosome instability and recurrent somatic copy number alterations (SCNA). Likewise, the highest grade of glioma, namely glioblastoma, showed a remarkably higher aneuploidy score than low-grade glioma across samples, which was consistent with previous findings that the prevalent increases in aneuploidy were detected at late stages of tumorigenesis in genetically engineered mouse models [92,93]. Of note, we observed the TRIP13 overexpression in lower-grade glioma with high-aneuploidy and measured a positive correlation between TRIP13 expression levels and aneuploidy score acquired from TCGA samples, but found no significant association between TRIP13 level and aneuploidy in GBM. Even if our results in LGG were consistent with previous findings that aneuploidy patterns might be driven by a specific gene that controls cell cycle and proliferation as well as indicative of inducing tumorigenesis [37,38,39], but the mechanism still remains unknown and it is suggested to be involved in potential heterogeneities between different grades of gliomas. Other studies have characterized the aneuploidy in tumor samples on the basis of aberrant expression of genes located on chromosomal regions. These identified signatures with top-ranked “CIN score” include TPX2, FOXM1, KIF20A, CCNB2, CDC20, AURKA, AURKB, NEK2, PRC1 and ZWINT [38,94]. Our study also observed that these signatures, which considered key components of cell-cycle and DNA damage regulation, were predicted to participate in TRIP13 co-expression networks. It reflected that the over-expressed TRIP13 may potentially participate in the molecular mechanism connecting the generation or maintenance of aneuploidy in glioma. It has also been found that the degree and presence of aneuploidy or related driver genes are associated with various aspects of tumorigenesis, such as cell proliferation, migration and immune evasion [34,95].

Recent clinical studies revealed that the degree of tumor aneuploidy is correlated with the markers of immune evasion, local immunity suppression and reduction of immunotherapeutic response [38,95]. On the other hand, other evidence suggested that aneuploidy is associated with the induction of some immune recognition of tumor cells [96]. Therefore, the activation or suppression of the immune system regulated by tumor aneuploidy function appeared to depend on the carcinogenic stage and the complicated microenvironment [34]. We used the differential gene expression analyses to identify 13 signature genes representative of immune cell subtypes. A further finding is that these immune signatures were downregulated both in low-grade glioma and GBM with TRIP13 overexpression (Figure 10I). Prior research demonstrated that the ratio of immune cell types could be used to reflect the fidelity of solid tumor microenvironment and the prediction of clinical outcome, highlighting tumor-infiltrating immune cells as playing a critical role in tumor development and control. For example, a high ratio of CD8^+^ T cells to regulatory T cells (Treg cells) in the tumor microenvironment of ovarian [97] and breast cancer [98] has been found to be associated with favorable outcomes attributed to Treg cells that mediate peripheral tolerance and immune homeostasis by repressing autoreactive T cells. Moreover, intratumoral accumulation and activation of CD4^+^FoxP3^+^ Treg has been considered the principal immune escape mechanism in gliomas [99]. Macrophages are also one of the major populations of tumor-infiltrating immune cells. M1 (classically activated) macrophages exhibit immuno-stimulatory and tumor cytotoxicity functions and are characterized by the production of pro-inflammatory cytokines [100]. In contrast, M2 (alternatively activated) macrophages have anti-inflammatory functions, involving production of interleukin 10 (IL-10) and transforming growth factor b (TGF-b) to suppress the anti-tumor immune response and promote angiogenesis for tumor progression [101]. The role of macrophages polarization (named M1/M2 ratio) has been used to reflect the function of tumor-associated macrophages (TAM) and a high M1/M2 ratio has been found to be correlated with better survival in gastric cancer patients [102]. However, our data suggested there is a marked association between TRIP13 expression and CD8^+^/Treg ratio in LGG and GBM, but neither differ between other immune cell type ratio (such as M1/M2 ratio), implying that multiple immune activated and suppressive mechanisms may be involved in the tumor microenvironment, and it is unlikely caused by differences in levels of individual genetic factor. Nevertheless, the interplay between various immune cell subtypes is more complex and dynamic in tumor environments. Our findings are the first to uncover the novel function of TRIP13 and suggest a new linkage between TRIP13 and immune components in low- and high-grade glioma. The detailed molecular mechanism about TRIP13 and immune cell infiltration need to be further elucidated.

## 4. Materials and Methods

### 4.1. Datasets, Glioma Samples and Survival Analysis

The gene expression data GSE16011 and corresponding clinical data used in this study were obtained from the GEO databases (https://www.ncbi.nlm.nih.gov/geo, accessed on 7 March 2020) and related article [103]. For the Chinese Glioma Genome Atlas (CGGA) cohort (http://www.cgga.org.cn, accessed on 12 March 2020), the mRNAseq_693, mRNAseq_325 and mRNA-array_301 were included in the study. Moreover, 624 glioma samples from TCGA (http://cancergenome.nih.gov/, accessed on 18 April 2020) were also used for validation. In this study, grades II and III of gliomas were defined as the lower-grade glioma (LGG), whereas grade IV was glioblastoma multiforme (GBM). RNA-seq data for specific tumor anatomic structure in GBM, identified by H&E staining, was based on the Ivy Glioblastoma Atlas Project (Ivy GAP) (https://glioblastoma.alleninstitute.org/, accessed on 4 May 2020). The Human Protein Atlas (https://www.proteinatlas.org/, accessed on 25 May 2020) database, which contains information on genes and corresponding high-resolution immunohistochemistry images, was used to examine TRIP13 expression in different tissues and the subcellular localization. The Kaplan–Meier survival analysis was performed to estimate the effect of TRIP13 expression on overall survival (OS) and progression-free survival (PFS) from TCGA, CGGA and GSE data. The curves were plotted by GraphPad Prism software and the hazard ratios and *p*-values were calculated.

### 4.2. Analysis of TRIP13 Expression in Human Glioma

TRIP13 transcriptome expression data for the TCGA (*n* = 156 glioblastoma specimens with 4 adjacent normal tissue specimens) were downloaded from the GlioVis data portal (http://gliovis.bioinfo.cnio.es/, accessed on 13 June 2020). We recruited the other two datasets to investigate TRIP13 expression in each glioma grade (1) GSE16011 array: grade I = 6, grade II = 23, grade III = 83, grade IV = 150; (2) CGGA: grade II = 138, grade III = 143, grade IV = 140. For estimating the TRIP13 expression with glioma stem cells, we utilized the GEO profile database (GDS3885 dataset) consisting of glioblastoma stem-like (GS) cell lines (*n* = 27), corresponding to glioblastoma primary tumors (*n* = 12) and GS neurospheres (*n* = 17) for further study.

### 4.3. RNA Isolation and Quantitative Real-Time Reverse Transcription-PCR

U87MG, LN229, GBM8401, U118MG and LNZ308 cells were maintained in Dulbecco’s modified Eagle’s medium (DMEM) and supplemented with 10% fetal bovine serum (FBS), penicillin and streptomycin. Total RNAs of glioma cell lines were extracted using the EasyPure Total RNA reagent (Bioman, NTPC, Taiwan) according to the manufacturer’s protocol. cDNAs were prepared using oligo dT, and MMLV Reverse transcriptase (Epicentre Biotechnologies, Madison, WI, USA). The normal brain cDNA was purchased from Origene Technologies (Rockville, MD, USA). Quantitative real-time PCR was performed with SYBR Green Master Mix on an IIIumina ECO™ Real-Time PCR system. Gene expression was measured by relative quantification method with normalization to GAPDH. The primer sequences were listed as follows: TRIP13 forward, 5′-ACTGTTGCACTTCACATTTTCCA-3′ and reverse, 5′-TCGAGGAGATGGGATTTGACT-3′ and GAPDH forward, 5′-CTTCATTGACCTCAACTAC-3′ and reverse, 5′-GCCATCCACAGTCTTCTG-3′. The amount of RNA was calculated using the 2^−ΔΔCt^ method.

### 4.4. Cell Lysate Preparation and Immunoblot Analysis

Cells were homogenized in lysis buffer (100 mM Tris-HCl, 150 mM NaCl, 0.1% SDS, and 1% Triton-X-100) at 4 °C and centrifuged to collect the supernatants. The normal brain lysates were obtained from Abcam. Equal protein quantities of the lysates were separated via 10% SDS-PAGE and then transferred to a polyvinylidene difluoride membrane (Bio-Rad Laboratories, Inc., Hercules, CA, USA). The primary antibodies for TRIP13 (Atlas Antibodies, STO, SWE) and GAPDH (Santa-Cruz, Dallas, TX, USA) were used. The protein bands were detected by enhanced chemiluminescence and imaged with X-ray film. The protein expression of GAPDH (Appendix A) and TRIP13 (Appendix A) was determined by Western bolt.

### 4.5. Identification of TRIP Co-Expressed Genes and Pathway and Network Analysis

We identified the TRIP13 co-expressed genes in glioma tumor by Pearson correlation analysis from GSE16011 array, CGGA301 RNA-seq and TCGA RNA-seq and GTEx from the Gene Expression Profiling Interactive Analysis (GEPIA) database (http://gepia2.cancer-pku.cn/#index, accessed on 20 May 2020). The listed genes correlated with expression of TRIP13 (Pearson R > 0.56) were subjected to construct by heat maps. The gene ontology (GO) and Kyoto Encyclopedia of Genes and Genomes (KEGG) analysis were performed from the Database for Annotation, Visualization and Integrated Discovery (DAVID) (https://david.ncifcrf.gov/, accessed on 5 July 2020) and KO-Based Annotation System (KOBAS) (http://kobas.cbi.pku.edu.cn, accessed on 18 July 2020). We uploaded the lists of TRIP13 co-expressed genes to Ingenuity Pathway Analysis^®^ Software (IPA from Ingenuity^®^: http://www.ingenuity.com, accessed on 24 August 2020) and performed prediction analysis for Diseases and Functions Annotation, canonical pathway and upstream regulators. The GeneMANIA prediction server (https://genemania.org/, accessed on 12 October 2020) was also used to predict the integrated network between TRIP13 and co-expressed genes.

### 4.6. DNA Methylation Modification Analysis

To examine the DNA methylation status of the target gene in cancer, we used MEXPRESS program (https://mexpress.be/, accessed on 25 November 2020), which visualized data for TCGA expression, DNA methylation status, clinical data and their relationships. We used this tool to investigate methylation status of the TRIP13 in low-grade glioma and glioblastoma. The information on TRIP13 methylation levels and TRIP13 mRNA expression was obtained from the LinkedOmics tool (http://www.linkedomics.org/admin.php, accessed on 2 December 2020) with a methylation450 assay. The KEGG pathway analysis in TRIP13 methylation was also performed by gene set enrichment analysis (GSEA) function in the LinkedOmics tool.

### 4.7. Identification of Micro RNAs and Circular RNAs Targeting TRIP13

We utilized the starBase program (http://starbase.sysu.edu.cn/index.php, accessed on 20 December 2020) underlying miRNA prediction database (miRanda, PITA) and LinkedOmics-miRNASeq data to identify the common miRNAs that target TRIP13 based on miRNA-mRNA interactomes data. Further analyses of the correlation between miRNAs and TRIP13 mRNA and predictions of circRNA were performed by this program. The expression profiles of micro RNAs in lower-grade glioma (*n* = 15) and glioblastoma (*n* = 10) were investigated by GSE112009 dataset. 

### 4.8. The Correlation Analysis between Tumor Aneuploidy and TRIP13

The cBioportal for Cancer Genomics (http://cbioportal.org, accessed on 12 January 2021) provide TCGA data to visualize various information of study origin, profiled for copy number alternations, aneuploidy score, overall survival and their relationships with each other. Here, we screened GBM (*n* = 592) and low-grade glioma (LGG) (*n* = 514) samples for serial analysis, including the correlation between aneuploidy score and survival time for TRIP13 mRNA expression. The approach of reactome pathway analysis was previously mentioned.

### 4.9. Immune-Related Analysis

We used previous data of TRIP13-negatively genes (r < −0.4) from analysis of LinkedOmics to derive a set of featured genes (*n* = 13) whose expressions determine the following immune cells subtypes: T cells, B cells, NK cells, dendritic cells, macrophages, granulocytes and monocytes (Table 1), for further analysis based on serial published literatures. To estimate the degree of the reduction in the expression of gene set specific for the corresponding immune cells, the average expression level of the gene set was determined by calculating the RSEM (RNA-Seq by Expectation-Maximization) value of each gene after log transformation and normalization from the merged TCGA data of low-grade glioma and GBM. We evaluated the percentage decrease in the expression of genes specific for immune cell type between high and low TRIP13 level tumors by the percentage difference in the average expression level. We also utilized the TISDIB database (Tumor Immune System Interactions Database, http://cis.hku.hk/TISIDB/index.php, accessed on 6 March 2021) and review literature [60,61,62,63] to derived a list of specific genes for CD8^+^ T cell, regulatory T (Treg) cell, M1 and M2 macrophage as well as the immune modulators such as cytokines (Appendix A). For calculating the ratios of CD8^+^ T cell-specific to Treg-specific genes, M1-specific to M2-specific genes and anti- to pro-tumor modulators, we calculated the ratio between the average expression levels (log2 transformed RSEM value) of each gene set for immune cell types. Finally, the Kaplan–Meier survival analysis was used to compare overall survival time between high and low ratio. The hazard ratios and *p*-values were calculated as the previous method.

### 4.10. Statistical Analysis

Student’s *t*-test or one-way analysis of variance (ANOVA) performed in GraphPad Prism 7 were used for statistical analysis of all mRNA expression experiments. Correlation analysis was evaluated using Pearson correlation. A *p*-value < 0.05 was considered to indicate significant differences. Corresponding significance levels are presented in the figure. The results are presented as the means ± standard deviation (SD) or as specified.

## 5. Conclusions

In summary, our data revealed that high TRIP13 expression was closely correlated with worse outcomes in low- and high-grade glioma. Lower-grade glioma with IDH-wild type expressed high TRIP13 levels in TCGA and CGGA data. We provided multi-level evidence to demonstrate the diverse function of TRIP13 in glioma. TRIP13 participated in several cancer-related pathways and the oncogenic potential in tumorigenesis may be likely involved in epigenetic regulation, miRNA-target interaction and the degree of aneuploidy, as well as the immune mediation in the tumor microenvironment. These findings are capable of providing a novel perspective to improve our understanding of the role of TRIP13 in glioma and thus warrant further evaluation of treatment strategies.

## Figures and Tables

**Figure 1 cancers-13-02338-f001:**
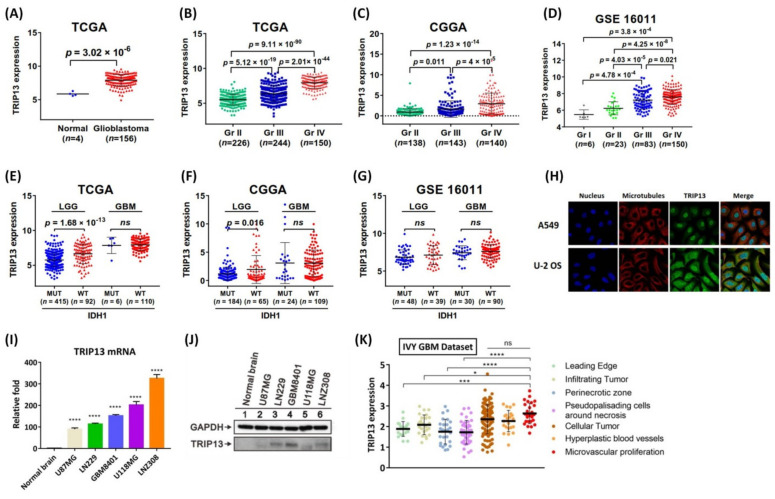
Systematically profiling the expression of TRIP13 in glioma. (**A**) The expression profiles of TRIP13 mRNA in glioblastoma and normal brain tissues based on TCGA are represented. Quantification data shows the expression levels of TRIP13 in gliomas from the TCGA, GSE16011 datasets, and CGGA stratified by the *IDH* status (**B**–**D**) and WHO grade (**E**–**G**). (**H**) The Human Protein Atlas database shows that TRIP13 was colocalized with nucleoplasm or microtubules in the cytoplasm of A549 and U-2 OS cells. (**I**,**J**) Validation of TRIP13 mRNA expression and protein production in U87MG, LN229, GBM8401, U118MG, and LNZ308 glioma cell lines in comparison with normal brain tissues. (**K**) TRIP13 expression was detected in the different subcellular location of GBM in the IVY GBM dataset. * *p*  <  0.05, *** *p* < 0.001, and **** *p*  <  0.0001 compared to the normal brain tissue group. n.s: not significant.

**Figure 2 cancers-13-02338-f002:**
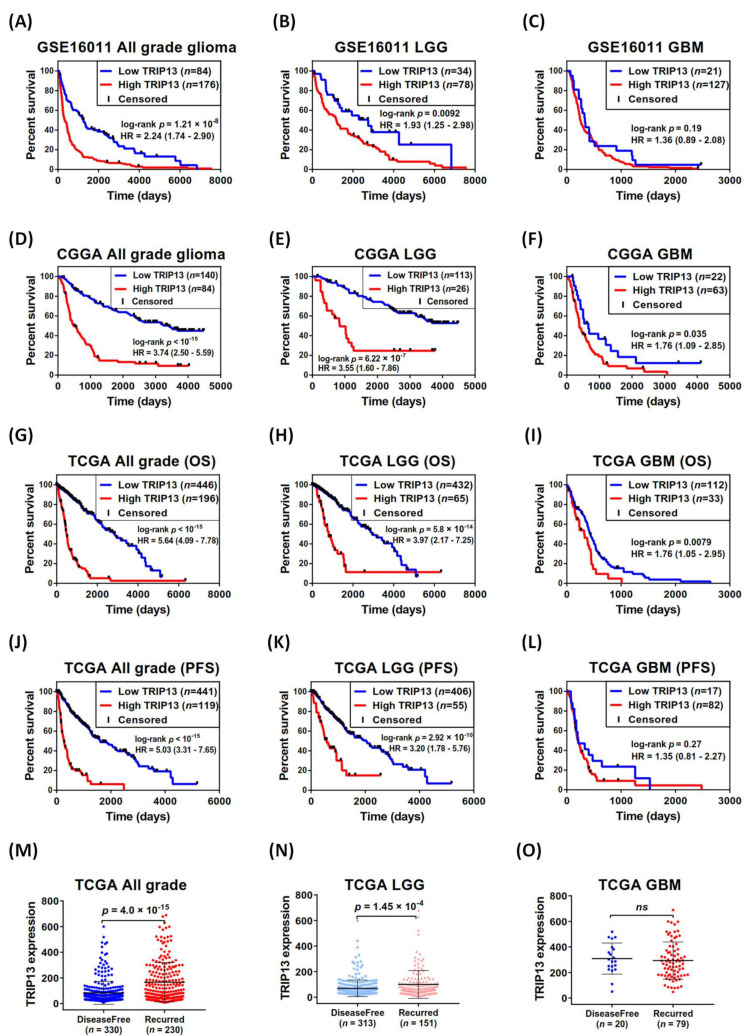
TRIP13 mRNA expression was related to clinical outcome in lower-grade glioma (LGG) and glioblastoma (GBM). (**A**–**C**) Kaplan–Meier estimates of overall survival for (**A**) all grade, (**B**) LGG and (**C**) GBM patients in the GSE16011 data. (**D**–**F**) Kaplan–Meier estimates of overall survival for (**D**) all grade, (**E**) LGG and (**F**) GBM patients in the CGGA. (**G**–**I**) Kaplan–Meier estimates of overall survival for (**G**) all grade, (**H**) LGG and (**I**) GBM patients in the TCGA. (**J**–**L**) Kaplan–Meier estimates of progression-free survival for (**J**) all grade, (**K**) LGG and (**L**) GBM patients in the TCGA. (**M**–**O**) TRIP13 expression individual in two distinct clinical status including disease free and recurred in (**M**) all grade, (**N**) LGG and (**O**) GBM patients (TCGA, Firehose Legacy). The *p*-values were calculated using the log-rank test. OS, overall survival. PFS, progression-free survival.

**Figure 3 cancers-13-02338-f003:**
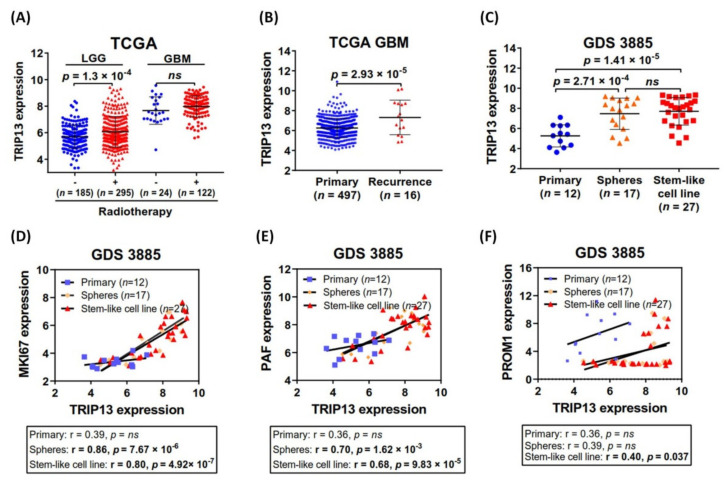
TRIP13 is preferentially expressed in recurrent glioblastoma and stem-like cell lines. (**A**) The expression profiles of TRIP13 mRNA in patients with or without radiotherapy based on TCGA lower-grade glioma (LGG) and glioblastoma (GBM) are represented. (**B**) Comparison of TRIP13 mRNA expression levels in primary and recurrent GBM in the TCGA dataset. (**C**) Comparison of TRIP13 mRNA expression levels in primary tumors (*n* = 12), GS neurospheres (*n* = 17) and glioblastoma stem-like cell lines (*n* = 27) in the GDS3885 dataset. Correlation analysis between TRIP13 and (**D**) MKI67, (**E**) PAF, and (**F**) PROM1 mRNA expression levels in primary tumors, neurospheres and glioblastoma stem-like cell lines in the GDS3885 dataset. r, Pearson correlation coefficient. The *p*-value indicates the significance of the correlation.

**Figure 4 cancers-13-02338-f004:**
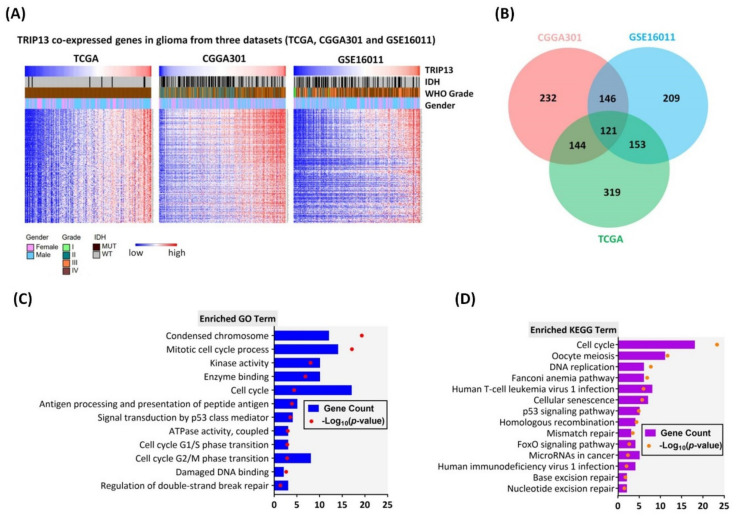
Functional enrichment analysis associated with TRIP13 expression level in human gliomas based on three datasets (TCGA, CGGA301 and GSE16011). (**A**) Heatmaps of clinicopathological parameters, TRIP13 co-expressed genes (R > 0.6) expression based on TRIP13 expression in glioma from TCGA, CGGA301 and GSE16011 array data. (**B**) Venn analysis of TRIP13 co-expressed genes. There were 121 overlapping genes positively associated (R > 0.6) with TRIP13 in TCGA, CGGA301 and GSE16011 array data. (**C**) GO and (**D**) KEGG pathway analysis of 121 overlapping genes in TCGA, CGGA301 and GSE16011 array data. CGGA, Chinese Glioma Genome Atlas; TCGA, The Cancer Genome Atlas; GO, Gene Ontology; Mut, mutant; WT, wild type.

**Figure 5 cancers-13-02338-f005:**
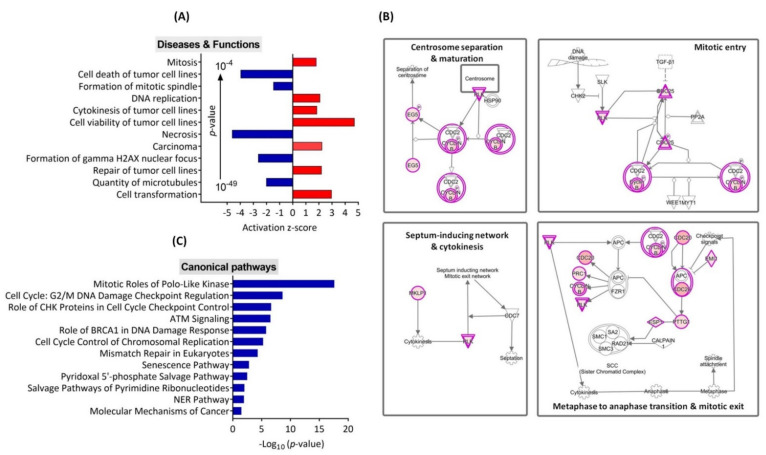
Identification of TRIP13 co-expressed genes functions and canonical pathway using IPA. (**A**) Ingenuity pathways analysis (IPA) Diseases and Functions Annotation. Bars with positive z-scores indicate that functional activity is increased (red color), whereas negative z-scores represent decreased activity (blue color). (|z-score| > 1.4, *p*-value  <  0.0001). (**B**) Mitotic roles of polo-like kinases. TRIP13 co-expressed genes (in red) are highly expressed in all mitotic pathways involving polo-like kinases (Plks). (**C**) IPA shows canonical pathways most significantly (*p* < 0.05) associated with TRIP13. The *p*-value for each pathway is representative of a blue bar and reported as the negative log of the *p*-value.

**Figure 6 cancers-13-02338-f006:**
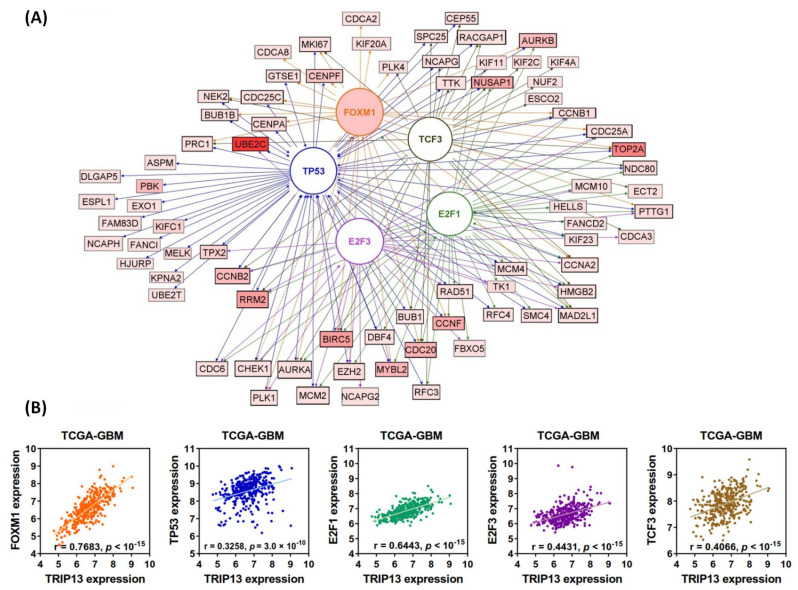
Network of interactions between the TRIP13 co-expressed genes and potential upstream regulators; confirmed relationships between TRIP13 and these regulators in TCGA-GBM database by correlation analysis. (**A**) The network was generated by the Path Designer tool of IPA. TP53, FOXM1, TCF1, E2F1 and E2F3 were the top regulators (different colors distinguish between them, with regulator–target gene connections by arrows/lines in the same colors), with high z-scores for activation. The circles represent proteins; squares indicate mRNA or genes. All mRNA for genes are displayed as expression values (low to high expression, light to dark red) from the TCGA database. (**B**) The scatter plot shows Pearson correlation (r) of TRIP13 expression with these transcription regulators.

**Figure 7 cancers-13-02338-f007:**
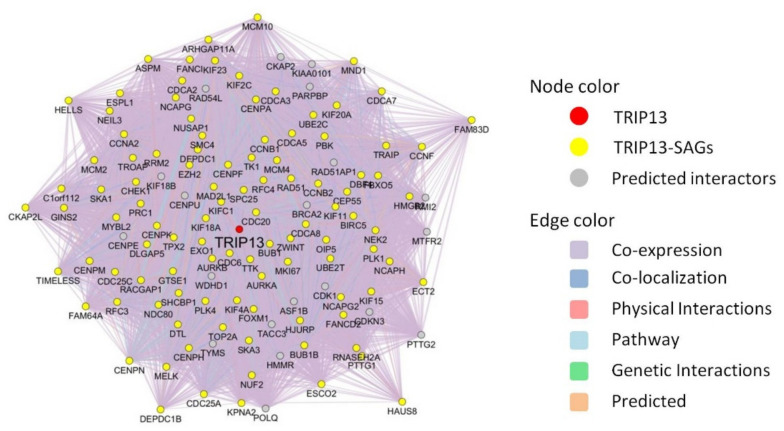
GeneMANIA network of TRIP13 and its co-expressed genes. The red node is the *TRIP13* gene, the yellow nodes are its co-expressed genes and the gray nodes are predicted interactors. The between-nodes edges indicate relationship types, including co-expression, co-localization, physical interaction, pathway, genetic interaction and predicted interaction, which are colored according to the legend. The thickness of edges represents interaction weight (i.e., strength of pairing relationships).

**Figure 8 cancers-13-02338-f008:**
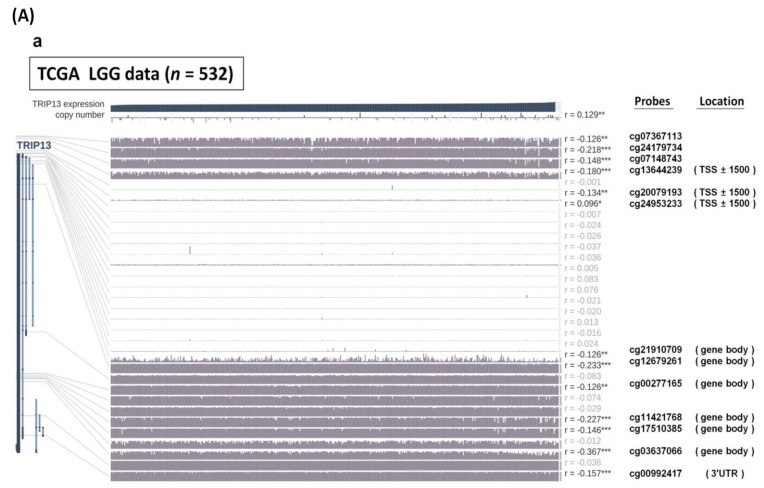
Associations among TRIP13 expression and DNA methylation and clinical characteristics in lower-grade glioma (LGG) and glioblastoma (GBM). (**A**) Visualization of the correlation between TRIP13 mRNA expression and DNA methylation in (**a**) TCGA-LGG and (**b**) TCGA-GBM cohort using the MEXPRESS program. The Pearson correlation coefficients (r) and *p*-value are indicated in the bottom panel. TSS, transcription start site; UTR, untranslated region. (**B**) The methylation status of *TRIP13* expression in low-grade glioma (*n* = 473) and GBM (*n* = 106) samples in TCGA. (**C**) Correlation analysis between methylation and gene expression levels of *TRIP13* of low-grade and high-grade glioma samples in TCGA. r, Pearson correlation coefficient of methylation and gene expression. P, *p*-value indicates the significance of the correlation. (**D**,**E**) Volcano plot of the distribution of all differentially expressed genes significantly associated with *TRIP13* methylation of LGG and GBM samples in TCGA. The red nodes represent upregulated differentially expressed genes; the blue nodes represent downregulated differentially expressed genes. The orange nodes represent the downregulated TRIP13 co-expressed genes (LGG, *n* = 118; GBM, *n* = 14). The gray dashed line indicates the −Log_10_ (*p*-value) cutoff value of less than 1.30 (means *p*-value < 0.05). (**F**) KEGG pathway analysis in *TRIP13* methylation using GSEA incorporated with LGG and GBM samples. Normalized enrichment scores and *p*-value corrected by FDR were calculated by GSEA. Only significantly enriched pathways (FDR adj. *p*-value < 0.05) are shown. Color represents the enrichment analysis results (blue, negative; red, positive). (**G**,**H**) Kaplan–Meier estimates of survival for LGG and GBM patients with hyper- or hypo-methylated *TRIP13* in TCGA. TCGA, The Cancer Genome Atlas; GBM, glioblastoma; GSEA, gene set enrichment analysis; KEGG, Kyoto Encyclopedia of Genes and Genomes; FDR, false discovery rate.

**Figure 9 cancers-13-02338-f009:**
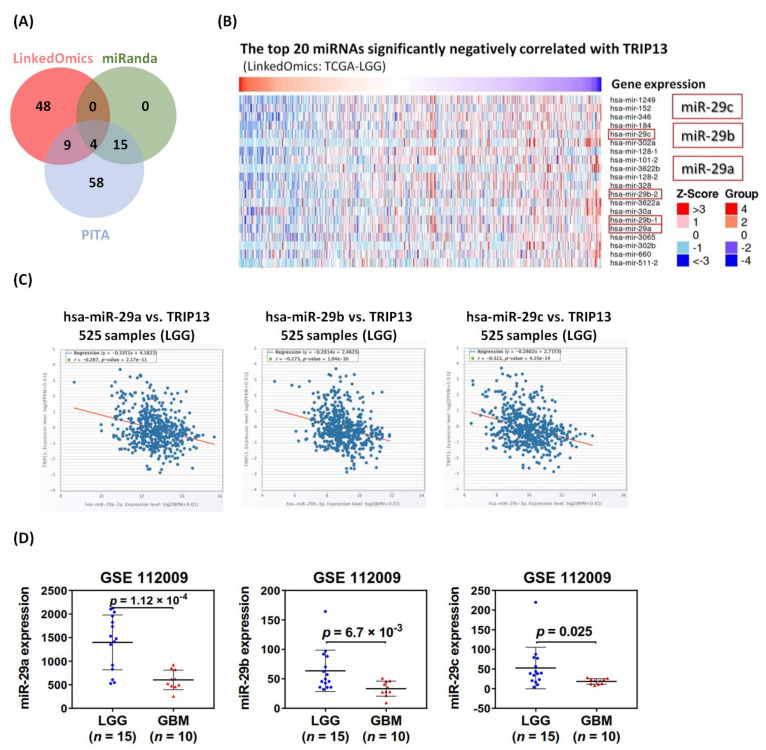
The miRNAs predicted to regulate TRIP13 based on lower-grade glioma (LGG) miRNA- sequence analysis. (**A**) Venn analysis of selected miRNAs predicted interactive with TRIP13. There were 4 overlapping miRNAs significantly negatively associated (r > −0.2) with TRIP13 in LinkedOmics, miRanda and PITA datasets. (**B**) Heatmaps of miRNAs negatively correlated with TRIP13 mRNA expression in the TCGA-LGG dataset (LinkedOmics). The miR-29 family (29a, 29b, 29c) are shown significantly inversed with TRIP13 based on 512 lower-grade glioma samples. (**C**) The starBase program demonstrated that TRIP13 mRNA expression was negatively correlated with the expression of miR-29 family (522 lower-grade glioma samples). (**D**) The expression profiles of miR-29 family (29a, 29b, 29c) mRNA were validated in LGG and GBM samples, respectively, in the GSE112009 dataset.

**Figure 10 cancers-13-02338-f010:**
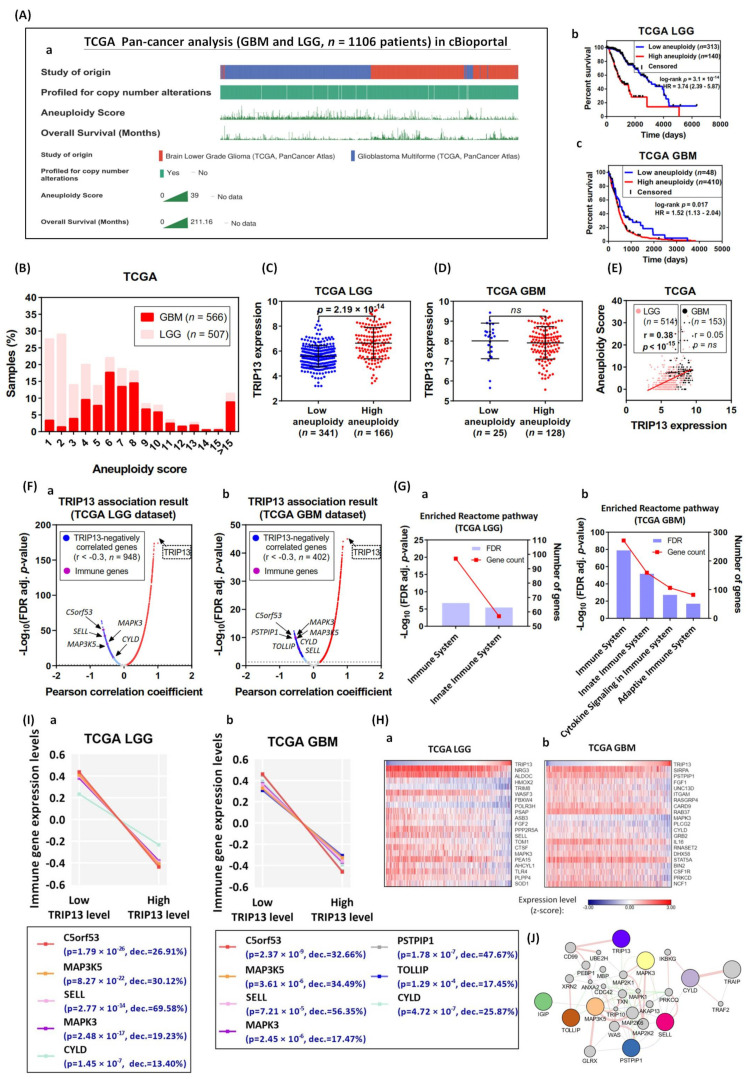
TRIP13 correlates with tumor aneuploidy and reduced immune expression signatures. (**A**) OncoPrint depicting somatic copy number alterations, aneuploidy score (reflecting the number of altered chromosome arms) (**a**) and overall survival time (**b**,**c**) using the cBioportal. The bottom-right “Kaplan–Meier curve” shows the overall survival time in TCGA-LGG (*n* = 443) and GBM (*n* = 458) cohorts stratified on the basis of high versus low aneuploidy score. (**B**) The plot shows the percentage of tumor samples in TCGA-LGG and GBM cohorts (*n* = 507 and *n* = 566, respectively) versus corresponding aneuploidy score which is determined by ABSOLUTE algorithm. (**C**,**D**) TRIP13 expression in low and high aneuploidy levels in TCGA LGG (*n* = 507) and GBM (*n* = 153) samples. (**E**) The correlation analysis between the transcriptional levels of TRIP13 and aneuploidy score for GBM (*n* = 153) and LGG (*n* = 514) tumor samples. r, Pearson correlation coefficient. The *p*-value indicates the significance of the correlation. (**F**) Volcano plot of the distribution of all differentially expressed genes significantly associated with TRIP13 of (**a**) LGG and (**b**) GBM samples in TCGA. The red nodes indicate upregulated differentially expressed genes; the blue nodes indicate downregulated differentially expressed genes. Among them, the 948 genes in LGG and 402 genes in GBM are significantly negative correlated (r < −0.3) and highlighted in deep blue, and several immune-related genes (e.g., *c5orf53, MAP3K5, SELL, CYLD, MAPK3* and *PSTPIP1*) are labeled on the plot. The gray dashed line indicates the –Log10 (*p*-value) cutoff value of less than 1.3 (means *p*-value < 0.05). (**G**) The Reactome pathway analysis reveals significant enrichment of negatively correlated genes with TRIP13 involved in major immune pathways in (**a**) LGG and (**b**) GBM. The top significant term is “immune system”. (**H**) Heat map demonstrating expression (z-score in RNA-seq by Expectation-Maximization) of the top 20 genes in the “immune system” term of (**F**) based on the Pearson correlation coefficient r in (**a**) LGG and (**b**) GBM. TCGA LGG and GBM samples are ordered by TRIP13 expression level. (**I**) The expression of genes specific for distinct types of immune cells (see Table 1) were evaluated in TCGA LGG and GBM with low or high TRIP13 levels. The legend illustrating gene symbols and expression pattern is obtained from the RNA-seq analysis (% dec., percent decrease). (**J**) The network of TRIP13 and immune-related genes based on (**I**). The blue node is TRIP13 gene, the other color nodes are immune-related genes, and the gray nodes are predicted interactors. The between-nodes edges indicate relationship types, including co-expression, co-localization, physical interaction and pathway which are colored according to the legend. The thickness of edges represents interaction weight (i.e., strength of pairing relationships).

**Figure 11 cancers-13-02338-f011:**
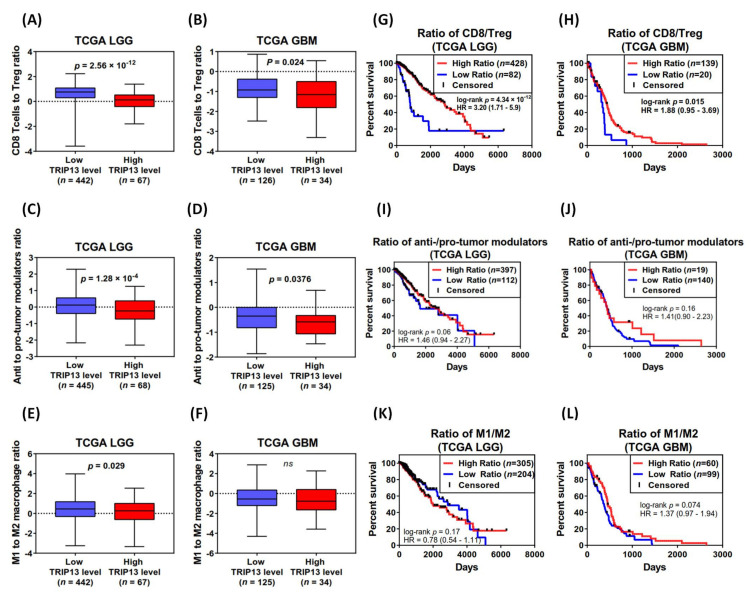
The ratio of immune cell type associated with TRIP13 expression and the corresponding Kaplan–Meier curves in lower-grade glioma (LGG) and glioblastoma (GBM) form TCGA. The boxplots displaying the gene expression ratio (RSEM normalization values) in LGG and GBM based on (**A**,**B**) CD8^+^ T cell-specific to Treg-specific genes, (**C**,**D**) anti-tumor to pro-tumor modulators specific genes and (**E**,**F**) M1 macrophage-specific genes to M2 macrophage-specific genes. Kaplan–Meier estimates of survival for high/low ratio of patients in TCGA LGG and GBM data based on (**G**,**H**) CD8^+^ T cell-specific to Treg-specific genes, (**I**,**J**) anti-tumor to pro-tumor modulators specific genes and (**K**,**L**) M1 macrophage-specific genes to M2 macrophage-specific genes.

**Table 1 cancers-13-02338-t001:** Immune genes negatively correlated with TRIP13 expression in TCGA GBM and LGG samples.

Gene	Description	*p*-Value	Pearson R	*p*-Value	Pearson R	Immune Cell Types	Refs
		LGG		GBM			
*C5orf53*	Chromosome 5 open reading frame 53	1.26 × 10^−53^	−0.61	1.11 × 10^−12^	−0.54	B cells, Dendritic cells	[42,43]
*MAP3K5*	Mitogen-activated protein kinase kinase kinase 5	7.04 × 10^−45^	−0.57	2.22 × 10^−8^	−0.43	T cells, Macrophages	[44,45]
*SELL*	Selectin L	4.83 × 10^−33^	−0.49	1.33 × 10^−6^	−0.38	T cells, NK cells, Granulocytes	[46,47,48]
*MAPK3*	Mitogen-activated protein kinase 3	3.8 × 10^−31^	−0.48	7.18 × 10^−9^	−0.45	T cells, Granulocytes	[49,50]
*CYLD*	CYLD Lysine 63 Deubiquitinase	8.61 × 10^−13^	−0.31	2.73 × 10^−8^	−0.43	T cells	[51]
*PSTPIP1*	Proline-serine-threonine phosphatase interacting protein 1	5.08 × 10^−11^	−0.28	3.56 × 10^−12^	−0.53	T cells, Granulocytes	[52,53]
*TOLLIP*	Toll interacting protein	2.18 × 10^−9^	−0.26	1.12 × 10^−6^	−0.38	Granulocytes, Monocytes	[54,55]

## Data Availability

All source data relating to this manuscript are available upon request.

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
