# Peer review of "Clinical Significance and Systematic Expression Analysis of the Thyroid Receptor Interacting Protein 13 (TRIP13) as Human Gliomas Biomarker"

_cancers, 2021, doi:10.3390/cancers13102338_

Round 1

Reviewer 1 Report

In their current work, Chen et al. use multiple publicly available glioma datasets (The Cancer Genome Atlas (TCGA), Chinese Glioma Genome Atlas (CGGA), and GSE316011) to investigate the expression, biological, and clinical significance of the Thyroid Receptor Interacting Protein 13 (TRIP13). Initially, the authors demonstrate an association between TRIP13 expression and glioma grade/IDH1 mutation status/recurrence/stemness/ methylation/aneuploidy/immune markers that portends poor survival. However, the data presented fails to convince the readers that TRIP13 has a role in glioma progression and survival.

Comments:
  1. In general, LGG patients have longer survival compared to HGG patients. However, throughout the manuscript, the authors combine the data from different glioma grades to establish a correlation between TRIP13 expression and several variables including, survival, radiation, enrichment analysis, aneuploidy, and immune infiltration (Figures 2, 3A, 4A, 8E, 8G, 9B, 10, and 11). To support their conclusion, the authors should separate the patient population based on the tumor grade and establish a correlation between TRIP13 expression and the different variables investigated.
  2. In Figure 1 the authors demonstrate a clear association between TRIP13 and glioma grade and TRIP13 expression. They also show an association between IDH mutational status and TRIP13 expression. The authors should include data to show if the association between TRIP13 and IDH1 mutation exists across different glioma grades.
  3. In Figure 1D, the authors should remove the single outlier from the IDH1-WT cohort and plot the chart with the same scale as 1B and 1C.
  4. In Figures 3D-3F, in addition to primary and stem cell lines, the authors should add the neurosphere samples for correlation analysis.
  5. In Figure 8C, the authors show an inverse correlation between promoter methylation and TRIP13 mRNA expression. However, they also demonstrate a positive correlation between the different DNMTs and TRIP13. It is challenging to reconcile the TRIP13 inverse correlation with promoter methylation and TRIP13 positive correlation with the different DNMTs.
  6. The Figure 9E data does not show any link between miR-29 and circRNAs in gliomas. The circRNA data show the authors hypothesis than establishing a role in TRIP13 regulation in gliomas. The circRNA information fits better in the discussion section than in the results section.
  7. The data and conclusion from Figures 10 and 11 are not valid unless they are separated by glioma grade as specified in comment 1.
  8. Since the authors establish an inverse correlation between TRIP13 mRNA expression and promoter methylation, the significance of using DNMT inhibitor for glioma treatment in the discussion section is confusing.
  9. In the methods section, the authors describe that “TRIP13 transcriptome expression data for the GSE-Glioma cohort (n = 582 cancerous specimens with 386 adjacent normal tissue specimens) were downloaded from the Gene Expression database of Normal and Tumor tissues 2 (GENT2) database (http://gent2.ap-pex.kr/gent2/).” The authors should include the data comparing TRIP13 expression between cancer (N=582, if applicable separate them by glioma grade) and adjacent normal tissue (N=386).
  10. To improve the clarity of the manuscript, the authors are recommended to make improvements in grammar and sentence formation.

Author Response

COMMENTS FOR THE AUTHOR:

Reviewer #1: In their current work, Chen et al. use multiple publicly available glioma datasets (The Cancer Genome Atlas (TCGA), Chinese Glioma Genome Atlas (CGGA), and GSE316011) to investigate the expression, biological, and clinical significance of the Thyroid Receptor Interacting Protein 13 (TRIP13). Initially, the authors demonstrate an association between TRIP13 expression and glioma grade/IDH1 mutation status/recurrence/stemness/ methylation/aneuploidy/immune markers that portends poor survival. However, the data presented fails to convince the readers that TRIP13 has a role in glioma progression and survival.

Response: We thank the Reviewer for these positive, supportive comments to improve the quality of this manuscript. We carefully conducted point-to-point response to the comments from Reviewer to address these issues.

Point 1: In general, LGG patients have longer survival compared to HGG patients. However, throughout the manuscript, the authors combine the data from different glioma grades to establish a correlation between TRIP13 expression and several variables including, survival, radiation, enrichment analysis, aneuploidy, and immune infiltration (Figures 2, 3A, 4A, 8E, 8G, 9B, 10, and 11). To support their conclusion, the authors should separate the patient population based on the tumor grade and establish a correlation between TRIP13 expression and the different variables investigated.

Response 1: We thank the Reviewer for this excellent suggestion. We have further separated the patient population based on the tumor grade and establish a correlation between TRIP13 expression and the different variables investigated. In this study, grades II and III of gliomas were defined as the lower-grade glioma (LGG), whereas grade IV was glioblastoma multiforme (GBM). (The sentences were added in section of Materials and Methods) please see Line 727~729 on Page 24. We have taken all the suggestions made by the reviewer and have the following changes.

  • Figure 2: We have investigated the correlation between TRIP13 expression and overall survival in LGG and GBM patients. We found that the LGG and GBM patients with high TRIP13 expression both had poor outcome in CGGA and TCGA. Please see 2A-O. Line 124~126 on Page 4. Besides, considering the original Figure 2C “Rembrandt survival analysis” study also based on TCGA GBM data to analysis, we have removed the figure. On the other hand, the original Figure 2G-H “Prognostic significance of TRIP13 in glioblastoma patients with or without TP53 mutations” described “a significant correlation was observed between of increased TRIP13 levels and poor overall survival in glioblastoma patients with TP53 mutations, compared to patients with wild-type TP53 phenotype”. When we review these section again and explored the correlation with TRIP13 mRNA level and TP53 status, the results indicated no statistical significance (p = 0.23, the figure as below). To avoid the ambiguity of the results, we have removed the figure and deleted the results and discussion regarding the issue. We also apologized for confusion we made.

  • Figure 3A: We have investigated the correlation between TRIP13 expression and radiation status in LGG and GBM patients. We found that the mRNA levels of TRIP13 increased in response to radiation (n = 417), compared to those without radiotherapy (n = 209) for LGG samples in TCGA (no statistically significant difference in GBM). We thought that radiosensitive role for TRIP13 may be different on grades of glioma. Please see 3A. Line 148~151 on Page 6

  • Figure 4A: We fully agree with Reviewer’s viewpoint and thank for this thoughtful comment to improve the quality of this manuscript. In fact, we have used the TCGA data, which is a landmark cancer genomics program, to perform the enrichment analysis of LGG and GBM by LinkedOmics program. We found that the top term of GO and KEGG pathways of LGG were almost similar with GBM (the data as below, the red color represent the overlapping term among TCGA-LGG, TCGA-GBM and original data).

According these results, we thought our data could support that major GO and KEGG pathways for TRIP13 expression between LGG and GBM TRIP13 would be no significant difference. We therefore decided to leave the figure as it. We sincerely hope that reviewer could approve our explanation.

  • Figure 8E: We have investigated the correlation between TRIP13 methylation levels and the differentially expressed genes (DEGs) in LGG and GBM, we used TCGA LGG and GBM methylation dataset to identify the 118 genes in LGG and 14 genes in GBM, and generated volcano plots to visualize the distribution of DEGs. To perform the ideal distribution analysis for LGG and GBM, respectively, the cutoff value (–Log10 (P-value)) has changed into 1.30 (means p-value < 0.05) for the purpose uniformly. The legend of figure 8D-E has been revised accordingly. Please see Line 344~348 on Page 14. We found that these DEGs were significantly negative correlated with TRIP13 methylation levels, suggesting the gene networks might be affected by the DNA methylation of TRIP13 gene in LGG and GBM. Please see 8D-E. on Page 14

  • Figure 8G: We have investigated the correlation between TRIP13 methylation level and overall survival in LGG and GBM patients. We found that LGG patients with hypermethylation predict better outcome, but not observed in GBM patients. We thought this result may reflect the fact regarding with the potential tumor heterogeneities on grading and other remaining unknown mechanisms in GBM. Please see Line 329~333 on Page 12.

  • Figure 9B: We thank the Reviewer for this excellent suggestion. We have tried to perform the TCGA GBM dataset (from LinkedOmics portal) to analysis for this purpose. Yet, we found that TCGA-GBM dataset did not include the miRNASeq data for further study. Of note, the sample cohort termed” TCGA GBM LGG” including 512 samples for miRNA study, which the sample size was same as termed” TCGA LGG” (the figure as below). (Figure as below) To clarify this, we reviewed the TCGA GBM LGG data actually only include the “LGG” samples for miRNA analysis. Besides, we have attempted to search for a long time for this purpose. Unfortunately, we have no access to this data and therefore are able to perform the requested analysis. Therefore, we expected to use the accessible TCGA LGG data to perform the miRNA analysis in this study. And the legend of figure 9B has been revised accordingly. Please see 9B. Line 375~382 on Page 15. We therefore decided to leave the figure as it. We sincerely hope that reviewer could approve our explanation.

In addition, in order to clearly present the expression of miR-29s expression in lower-grade glioma, we preferred used the GSE112009 data, which performed the miRNA expression between lower-grade glioma (n = 15) and glioblastoma (n = 10), rather than the original analysis between “normal brains vs. GBM”. Therefore, we decided to remove the original Fig. 9D, and added the new Fig. 9D as below. The legend of Fig 9D has been revised accordingly. Please see Fig. 9D. Line 381~382 on Page 15.

  • Figure 10: Initially, we have investigated the correlation between aneuploidy level and survival in LGG and GBM We found that patients with high-aneuploidy had markedly worse survival compared to low-aneuploidy patients in LGG and GBM. (LGG: median survival, 1091 days for high groups versus 3156 days for low groups, GBM: median survival, 434 days for high groups versus 587 days for low groups) (Fig 10A, a-b). Then, we further investigated the correlation between TRIP13 expression and aneuploidy level in LGG and GBM patients. We found that LGG with high levels of aneuploidy showed significant elevated expression of TRIP13 (Fig 11C), and the positive correlation (r = 0.38, P < E-15) between TRIP13 expression and aneuploidy score (Fig 10E). However, we did not observe significant difference with respect to the correlation between TRIP13 level and aneuploidy in GBM samples (Figure 13D-E) Please see Fig 10. Line 419~423 on Page 16. We thought these results may attribute to some unknown mechanism remained and suggested involved potential heterogeneities between different grades of gliomas. Next, we also have investigated the correlation between TRIP13 exprssion and the negatively differentially expressed genes (DEGs) in LGG and GBM, and generated volcano plots to visualize the distribution of DEGs. To perform the ideal distribution analysis for LGG and GBM, respectively (Please see Fig 10F, Line 477~483 on Page 19), the cutoff value (–Log10 (P-value)) has also changed into 1.30 (means p-value < 0.05) and Pearson correlation (r) changed into < -0.3 for the purpose uniformly. The legend of figure 10F has been revised accordingly. According the volcano plots, several immune-related genes in LGG (n = 5) and GBM (n = 7) have been again indentified and revised as Table I (Page 19) Then, we performed “Enriched Reactome pathway analysis” (Fig 10G) and “Heat map” (Fig 10H) in LGG and GBM. Finally, we have investigated the correlation between TRIP13 expression and the identified immune genes expression in LGG and GBM, respectively. The results indicated that both in LGG and GBM, overall expression of immune genes showed markedly reduction in tumors with high TRIP13 levels relative to low TRIP13 levels tumors (LGG: P = E-07~E-26, GBM: P = E-04~E-09) Please see Fig. 10I. on Page 18. The network of seven immune-related genes and TRIP13 also performed and found that CD99 was reported physical interactive with TRIP13 (edge weight 6.33). Please see Fig. 10J. on Page 18.

  • Figure 11: We thank the Reviewer for this excellent suggestion. We have investigated the correlation between three immune cells type (CD8+ / Treg, anti-inflammatory modulators / pro-inflammatory modulators, and Macrophage 1 / Macrophage 2) ratio and TRIP13 expression as well as clinical outcome in LGG and GBM The results indicated that in high TRIP13 level tumors, the ratio between mRNA levels of CD8+ /Treg was notably reduced, compared to low TRIP13 level tumors (Fig 11A-B, Line 507~510 on Page 19~20) as well as have the worsen clinical outcome in LGG and GBM, but we did not observe the significant difference in the ratio of anti-/pro-tumor modulators, nor the ratio of M1/M2 (Fig 11G-H, Line 518~521 on Page 20). Based on these results, we considered that TRIP13 may play a potential role in the regulation of CD8+ and Treg cells in the tumor microenvironment between different grades of glioma. (Line 521~523 on Page 20)

Point 2: In Figure 1 the authors demonstrate a clear association between TRIP13 and glioma grade and TRIP13 expression. They also show an association between IDH mutational status and TRIP13 expression. The authors should include data to show if the association between TRIP13 and IDH1 mutation exists across different glioma grades.

Response 2: We thank the Reviewer for reminding us this important issue. In response to the reviewer’s suggestion, we have investigated the correlation between TRIP13 expression and IDH mutation in LGG and GBM patients. The results indicated the TCGA analysis exhibited that TRIP13 expression was higher in the isocitrate dehydrogenase 1 wild-type (IDH1-WT) group than in the IDH1 mutation (IDH1-MUT) group in lower-grade glioma (LGG), but not significant in primary GBM. The above results were validated from CGGA database but neither differs in LGG and GBM in GSE16011 array cohort.

Then, we have attempted to offer the explanation for this issue in the discussion section. These sentence read as: “We have Several studies showed that IDH1 mutation are very frequent in secondary (>80%) but very rare in primary glioblastoma (< 5%). IDH1 mutation is a definitive diagnostic molecular marker in secondary compared to primary glioblastoma. Hence, the study was limited for TRIP13 expression analysis in primary glioblastoma to explore the relation with IDH1”. And we added new references (reference 65-67). Please see Line 552~557 on Page 21.

Point 3: In Figure 1D, the authors should remove the single outlier from the IDH1-WT cohort and plot the chart with the same scale as 1B and 1C.

Response 3: We thank the Reviewer for the careful check our data. We have exchanged the Fig 1B-D with Fig 1E-G for more logical presentation (namely firstly present “different grade glioma” data, then present “IDH mutation “data). Therefore, we have removed the single outlier from the IDH1-WT cohort and plot the chart with the same scale as Fig 1E-F. Please see Fig. 1E-G.

Point 4: In Figures 3D-3F, in addition to primary and stem cell lines, the authors should add the neurosphere samples for correlation analysis.

Response 4: We thank the Reviewer for this excellent suggestion. We have added the neurosphere samples for correlation analysis. The legend of figure 3D-F has been revised accordingly. We observed that neurospheres samples performed high correlation with MKI67 and PAF, but no statistical significance with PROM1. Please see Fig 3D-F, Line 173~174 on Page 6.

Point 5: In Figure 8C, the authors show an inverse correlation between promoter methylation and TRIP13 mRNA expression. However, they also demonstrate a positive correlation between the different DNMTs and TRIP13. It is challenging to reconcile the TRIP13 inverse correlation with promoter methylation and TRIP13 positive correlation with the different DNMTs.

Response 5: We fully agree with the Reviewer’s viewpoints and have reviewed related articles regarding the correlation between DNMTs and gene methylated status. Therefore, we have investigated the correlation between TRIP13 methylated and DNMTs expression by using TCGA methylation data from LinkedOmics tool. The results indicated that TRIP13 methylated level had no correlation between DNMTs in LGG and GBM (as below), suggesting that there are other methylation mechanisms remained to elucify.

<TCGA-GBM>

<TCGA-LGG>

Thus, we apologize for the confusion we made. To avoid the confusion, we have removed the section of DNMTs and state our findings to focus on the correlation between TRIP13 methylation and other characteristics in lower-grade glioma and glioblastoma. Please see Line 288~333 on Page 12.

Point 6: The Figure 9E data does not show any link between miR-29 and circRNAs in gliomas. The circRNA data show the authors hypothesis than establishing a role in TRIP13 regulation in gliomas. The circRNA information fits better in the discussion section than in the results section.

Response 6: We thank the Reviewer for this excellent suggestion. We have moved the circRNA information to the discussion section. And we also reported the network between miRNA, circRNA and TRIP13 in the Supplementary Materials Fig.4. Please see Line 646~650 on Page 23.

Point 7: The data and conclusion from Figures 10 and 11 are not valid unless they are separated by glioma grade as specified in comment 1.

Response 7: We thank the Reviewer for this excellent suggestion. In response to the reviewer’s suggestion, we have taken all the suggestions made by the reviewer and have made the following analysis. (as response 1.)

  • Figure 10: Figure 10: Initially, we have investigated the correlation between aneuploidy level and survival in LGG and GBM We found that patients with high-aneuploidy had markedly worse survival compared to low-aneuploidy patients in LGG and GBM. (LGG: median survival, 1091 days for high groups versus 3156 days for low groups, GBM: median survival, 434 days for high groups versus 587 days for low groups) (Fig 10A, a-b). Then, we further investigated the correlation between TRIP13 expression and aneuploidy level in LGG and GBM patients. We found that LGG with high levels of aneuploidy showed significant elevated expression of TRIP13 (Fig 11C), and the positive correlation (r = 0.38, P < E-15) between TRIP13 expression and aneuploidy score (Fig 10E). However, we did not observe significant difference with respect to the correlation between TRIP13 level and aneuploidy in GBM samples (Figure 13D-E) Please see Fig. 10. Line 419~423 on Page 16. We thought these results may attribute to some unknown mechanism remained and suggested involved potential heterogeneities between different grades of gliomas. Next, we also have investigated the correlation between TRIP13 exprssion and the negatively differentially expressed genes (DEGs) in LGG and GBM, and generated volcano plots to visualize the distribution of DEGs. To perform the ideal distribution analysis for LGG and GBM, respectively (Please see Fig 10F, Line 477~483 on Page 19), the cutoff value (–Log10 (P-value)) has also changed into 1.30 (means p-value < 0.05) and Pearson correlation (r) changed into < -0.3 for the purpose uniformly. The legend of figure 10F has been revised accordingly. According the volcano plots, several immune-related genes in LGG (n= 5) and GBM (n= 7) have been indentified and listed at Table I (Page 19) Then, we performed “Enriched Reactome pathway analysis” (Fig 10G) and “Heat map” (Fig 10H) in LGG and GBM. Finally, we have investigated the correlation between TRIP13 expression and the identified immune genes expression in LGG and GBM, respectively. The results indicated that both in LGG and GBM, overall expression of immune genes showed markedly reduction in tumors with high TRIP13 levels relative to low TRIP13 levels tumors (LGG: P = E-07~E-26, GBM: P = E-04~E-09) Please see Fig. 10I. on Page 18. The network of seven immune-related genes and TRIP13 also performed and found that CD99 was reported physical interactive with TRIP13 (edge weight 6.33). Please see Fig. 10J. on Page 18.

  • Figure 11: we investigated the correlation between three immune cells type (CD8+ / Treg, anti-inflammatory modulators / pro-inflammatory modulators, and Macrophage 1 / Macrophage 2) ratio and TRIP13 expression as well as clinical outcome in LGG and GBM patients. The results indicated that in high TRIP13 level tumors, the ratio between mRNA levels of CD8+ /Treg was notably reduced, compared to low TRIP13 level tumors (Fig 11A-B, Line 507~510 on Page 20) as well as have the worsen clinical outcome in LGG and GBM, but we did not observe the significant difference in the ratio of anti-/pro-tumor modulators, nor the ratio of M1/M2 (Fig 11G-H, Line 518~521 on Page 20). Based on these results, we considered that TRIP13 may play a potential role in the regulation of CD8+ and Treg cells in the tumor microenvironment between different grades of glioma. (Line 521~523 on Page 20)

Point 8: Since the authors establish an inverse correlation between TRIP13 mRNA expression and promoter methylation, the significance of using DNMT inhibitor for glioma treatment in the discussion section is confusing.

Response 8: We thank the Reviewer for this excellent suggestion and fully agree with the reviewer’s viewpoints. Since we did not observe the correlation between TRIP13 methylated status and DNMTs, it is reasonable that removed the information about DNMT inhibitor for glioma treatment in this study. To avoid the confusion, we have removed this section in the discussion. And we apologize for the confusion we made.

Point 9: In the methods section, the authors describe that “TRIP13 transcriptome expression data for the GSE-Glioma cohort (n = 582 cancerous specimens with 386 adjacent normal tissue specimens) were downloaded from the Gene Expression database of Normal and Tumor tissues 2 (GENT2) database (http://gent2.ap-pex.kr/gent2/).” The authors should include the data comparing TRIP13 expression between cancer (N=582, if applicable separate them by glioma grade) and adjacent normal tissue (N=386).

Response 9: We thank the Reviewer for this excellent suggestion. In fact, the Fig 1A was analysis in TCGA data, but not in GSE-glioma cohort. To avoid the misleading, we have revised our statements in Materials and Methods section. The sentence read as: “TRIP13 transcriptome expression data for the TCGA (n = 156 glioblastoma specimens with 4 adjacent normal tissue specimens) were downloaded from GlioVis data portal (http://gliovis.bioinfo.cnio.es/).” And we apologize for error we made. Please see Line 740~742 on Page 24)

Point 10: To improve the clarity of the manuscript, the authors are recommended to make improvements in grammar and sentence formation.

Response 10: We thank the Reviewer for this excellent suggestion. We tried our best to improve the manuscript and made some changes in the manuscript in a more precise and concise way. These changes will not influence the content and framework of the paper. And here the Figures / Tables of changes marked in red in revised paper.

Finally, we appreciate for Reviewer’s warm work earnestly, and hope that the correction will meet with approval.

Reviewer 2 Report

The paper by Chen et al., described the possibility to use the thyroid receptor-interacting protein 13 (TRIP13) as a poor prognostic marker of GBM with wild-type IDH1. The authors performed an impressive bioinformatic and interactome analysis, between different grades of GBM, using different data banks such as TCGA. In particular, the data presented show a correlation between the expression of TRIP13 and patients with p53 and IDH mutation. Moreover, the authors suggest the involvement of TRIP13 in different oncogenic pathways such as DNA-methylation, miRNA, aneuploidy, and microenvironment-immune system modulation. This is a very interesting study, and well carried out, however, it seems that TRIP13 lacks a really specific effect to be an oncogenic factor, although the assumptions made by the authors are convincing. This is more a computational paper than an experimental one.

Author Response

COMMENTS FOR THE AUTHOR:

Reviewer #2: The paper by Chen et al., described the possibility to use the thyroid receptor-interacting protein 13 (TRIP13) as a poor prognostic marker of GBM with wild-type IDH1. The authors performed an impressive bioinformatic and interactome analysis, between different grades of GBM, using different data banks such as TCGA. In particular, the data presented show a correlation between the expression of TRIP13 and patients with p53 and IDH mutation. Moreover, the authors suggest the involvement of TRIP13 in different oncogenic pathways such as DNA-methylation, miRNA, aneuploidy, and microenvironment-immune system modulation. This is a very interesting study, and well carried out, however, it seems that TRIP13 lacks a really specific effect to be an oncogenic factor, although the assumptions made by the authors are convincing. This is more a computational paper than an experimental one.

Response: We thank the Reviewer for these positive, supportive comments to improve the quality of this manuscript. We carefully conducted point-to-point response to the comments from Reviewer to address these issues.

Point 1: Why did the authors not try to modulate the immune response by in vitro experiments (co-culture)?

  • Response 1: We thank the Reviewer for this thoughtful comment to improve the quality of this manuscript. We agree that in vitro experiments are important line of study. We have now acknowledged this and suggested it as a topic for further research in the discussion section. Please see Line 714~719 on Page 24.

Point 2: Did the authors analyze and/or found in GBM specimens some TRIP13 mutation?

Response 2: We thank the Reviewer for this excellent suggestion. We got glioblastoma with the TRIP13 mutation case. Here we demonstrated "TCGA-19-5956-01". The patient was a 53-year-old female diagnosed with glioblastoma with three missense mutations at p.Q194H, p.Y206C, and p.V366M in chromosome 5. The histology showed as the following picture and revealed increased cellularity, composed of neoplastic astrocytes with giant cellular morphology (arrowhead) and multi-nuclei. Area of endothelial proliferation (arrow) and tumor necrosis (asterisk) are also identified. Combine the above histopathologic findings, and it was a case indicative of glioblastoma, a giant-cell histological variant, with TRIP13 mutation.

Figure. A case of glioblastoma, a giant-cell histological variant, with TRIP13 mutation. The scale bar is 100 µm.

Point 3: To define TRIP 13 as an oncogenic factor it seems too strong at the moment.

Response 3: We thank the comment from the Reviewer to adequately define TRIP13. Accordingly, we have revised the manuscript in simple summary, discussion and conclusion section. The sentence read as:

  • The study aimed to provide comprehensive information about the oncogenic potential of TRIP13 in clinical significance for gliomas.(Please see: Line 20 on Page 1, simple summary section)
  • The TCGA data also present upregulation of TRIP13 displayed higher recurrence rates, overall survival and progression-free survival (PFS). Above finding support the potential oncogenic role of TRIP13 in human gliomas, and may serve as an indicator for patient with IDH mutations(Please see: Line 560-563 on Page 21, discussion section)
  • TRIP13 participated in several cancer-related pathways and the oncogenic potential in tumorigenesis may be likely involved in the epigenetic regulation, miRNA-target interaction, the degree of aneuploidy as well as the immune mediation in tumor microenvironment.” (Please see: Line 843-846 on Page 27, conclusion section)

We will take the Reviewer’s suggestion in our follow-up study.

Point 4: There are some text format errors throughout the manuscript.

Response 4: We thank the comment from the Reviewer. We have carefully to check and correct the text format errors throughout the manuscript and revised all references to follow the MDPI layout style. Please see Line 886~1111 on Page 27~33.

Finally, we appreciate for Reviewer’s warm work earnestly, and hope that the correction will meet with approval.

Round 2

Reviewer 1 Report

 None